# A biochemical network controlling basal myosin oscillation

Xiang Qin [1,2,3], Edouard Hannezo [4,5], Thomas Mangeat[1,2], Chang Liu[1,2,6], Pralay Majumder[7], Jiaying Liu[1,2], Valerie Choesmel-Cadamuro[1,2], Jocelyn A. McDonald[8], Yiyao Liu[3], Bin Yi[6] & Xiaobo Wang [1,2]

The actomyosin cytoskeleton, a key stress-producing unit in epithelial cells, oscillates spontaneously in a wide variety of systems. Although much of the signal cascade regulating myosin activity has been characterized, the origin of such oscillatory behavior is still unclear. Here, we show that basal myosin II oscillation in *Drosophila* ovarian epithelium is not controlled by actomyosin cortical tension, but instead relies on a biochemical oscillator involving ROCK and myosin phosphatase. Key to this oscillation is a diffusive ROCK flow, linking junctional Rho1 to medial actomyosin cortex, and dynamically maintained by a self-activation loop reliant on ROCK kinase activity. In response to the resulting myosin II recruitment, myosin phosphatase is locally enriched and shuts off ROCK and myosin II signals. Coupling *Drosophila* genetics, live imaging, modeling, and optogenetics, we uncover an intrinsic biochemical oscillator at the core of myosin II regulatory network, shedding light on the spatio-temporal dynamics of force generation.

[1] Université de Toulouse, UPS, F-31062 Toulouse, France. [2] CNRS, LBCMCP, F-31062 Toulouse, France. [3] Department of Biophysics, School of Life Science and Technology, University of Electronic Science and Technology of China, Chengdu, 610054 Sichuan, P. R. China. [4] The Wellcome Trust/Cancer Research UK Gurdon Institute, University of Cambridge, Cambridge CB2 1QN, UK. [5] Institute of Science and Technology Austria, Am Campus 1, A-3400 Klosterneuburg, Austria. [6] Department of Anesthesia, Southwest Hospital, Third Military Medical University, Chongqing 400038, P.R. China. [7] Department of Life Sciences, Presidency University, Kolkata 700073, India. [8] Division of Biology, Kansas State University, Manhattan, KS 66506, USA. These authors contributed equally: Xiang Qin, Edouard Hannezo. Correspondence and requests for materials should be addressed to Y.L. (email: liuyiyao@uestc.edu.cn) or to B.Y. (email: yibin1974@163.com) or to X.W. (email: xiaobo.wang@univ-tlse3.fr)

Tissue morphogenesis, the acquisition of the three-dimensional (3D) shape of tissues and organs, is driven by biological forces that are typically generated by molecular motors, such as non-muscle myosin II (Myo-II) pulling filamentous actin (F-actin)[1,2]. The activity of the Myo-II molecule, composed of two heavy chains (MHC), two regulatory light chains (MRLC), and two essential light chains, is regulated by dynamic phosphorylation and dephosphorylation of MRLC[3,4]. Phosphorylation and dephosphorylation of MRLC, mainly by *Rho*-associated protein kinase (ROCK) and myosin light-chain phosphatase (MLCP), respectively, govern an active or inactive state of the Myo-II molecule, thus controlling both its interaction with actin filaments and the generation of contractile force[4,5].

Actomyosin networks are very dynamic and often undergo cycles or pulses of assembly and disassembly[1]. Pulses of actomyosin networks have been observed in several *Drosophila* tissue morphogenesis processes, including apical constrictions of invaginating mesoderm cells and delaminating neuroblasts, ectoderm cell intercalation during germ band extension, follicle cells during oogenesis, and amnioserosa cells during dorsal closure, but also during *Xenopus* gastrulation and mouse embryo compaction[6–17]. Several competing explanations for this phenomenon have been put forward, including mechanosensitive assembly[10], balance between contraction forces and turnover[11], or phosphorylation–dephosphorylation cycles of MRLC[18]. Consistent with a control via phosphorylation–dephosphorylation cycles, spatio-temporal correlation between ROCK-MLCP and Myo-II has been observed in various morphogenetic processes[18–22]. However, how such cycles are generated and regulated is still unclear.

Basal actomyosin pulses, also essential for tissue morphogenesis, occur at the basal domain of ovarian epithelial follicle cells during *Drosophila* oogenesis[9], a biological process to give rise to mature eggs. This basal actomyosin oscillation is thought to create an anisotropic mechanical force, along the dorsal-ventral (DV) axis, in order to constrict the underlying tissue and thus eventually give the mature egg its characteristic anterior–posterior (AP) elongation[9]. Studies[9,19,23] have demonstrated that Rho1 and ROCK positively control basal Myo-II intensity, while Flapwing (Flw), one catalytic subunit of MLCP, negatively governs both basal Myo-II intensity and the initiation of basal Myo-II oscillation.

Here, through a combination of time-lapse live cell imaging, genetic, and optogenetic tools, and a physical based reaction diffusion modeling, we demonstrate that basal Myo-II oscillation is controlled by a biochemical oscillatory network involving spatio-temporal patterns of ROCK and MLCP, giving rise to activator/inhibitor dynamics.

## Results

### Basal shift of Myo-II regulatory network during elongation.

Basal Myo-II oscillation starts from the developing early stage 9 and it lasts until the end of stage 10B during *Drosophila* oogenesis[9]. During these periods, Myo-II displays a prominent shift from the apical to basal domain. Since ROCK and myosin-binding subunit (MBS, the regulatory subunit of MLCP[24]) are well known activator and inhibitor of Myo-II phosphorylation and its mediated contraction, we assessed the spatio-temporal distribution of both proteins in follicle cells. Consistent with Myo-II patterns, both ROCK and MBS proteins display prominent and gradual shifts from apical to basal domain from early stage 9 to stage 10B (Fig. 1a–d). As basal Myo-II activity plays a critical role in controlling tissue elongation[9], we tested whether tissue elongation would be affected upon changes of ROCK or MBS activity. Firstly, we confirmed the knockdown efficiency of

MBS RNAi in follicle cells (Supplementary Fig. 1a, b). Consistent with their predicted effect on Myo-II activity, the activation of ROCK (by overexpression of ROCK active form) and the inhibition of MBS (by MBS RNAi expression) both significantly enhance the elongation of egg chambers, while the inhibition of ROCK (by ROCK RNAi expression) and the activation of MBS (by overexpressing MBS active form[25]) both result in rounder egg chambers (Fig. 1e, f). These results indicate that ROCK and MBS are involved in regulating Myo-II activity and governing tissue elongation. We therefore asked whether ROCK, MBS, and other Myo-II regulators were involved in the oscillatory properties of Myo-II.

### Oscillatory phase shift within Myo-II regulatory network.

We started by assessing the spatio-temporal dynamics of the Rho1–ROCK–MLCP signaling cascade. Firstly, we noticed that Rho1 intensity and activity are strongly distributed at basal junctional membrane, with almost undetectable signal at basal medial cortex, where oscillating Myo-II is present (Fig. 2a, c and ref. [23]). In addition, we did not detect a prominent pulsatile pattern of basal membrane Rho1 intensity and activity, during periods of Myo-II pulses (Fig. 2b, d, k and Supplementary Movie 1). We next assessed the dynamics of ROCK and MLCP signals. Basal ROCK levels oscillate with the same period as basal Myo-II levels, but preceding Myo-II by nearly 1 min (Fig. 2e, f, j, l and Supplementary Movie 2). Next, we assessed the dynamic correlation between Myo-II and MLCP. Here, we tracked the spatio-temporal dynamics of MBS and Flw, the respective regulatory and catalytic subunits of MLCP[26]. To validate whether our MBS fluorescence reporters reflect with precision the endogenous MBS signals during basal Myo-II oscillation, we compared MBS fluorescence reporters and MBS antibody staining at different stages of Myo-II oscillations, and found that MBS reporters correlated strongly with MBS antibody stainings (Supplementary Fig. 1c–i). Importantly, although the total MBS protein levels per basal area remain constant throughout Myo-II oscillations, MBS displays periodic cycles of redistribution towards the medial pool where Myo-II is located, with the same period, but prominently trailing behind by almost 1 min (Supplementary Fig. 1j, k, 2a–c, g, i and Supplementary Movie 3). Similar to the spatio-temporal dynamics of MBS signals, endogenous Flw signals, detected by the protein-trap line *flw-YFP-159*[27], are spatially colocalized with, and temporally synchronized, with MBS signals (Supplementary Fig. 2d–f, h, i and Supplementary Movie 4). Thus, Flw also displays periodic cycles of redistribution toward the medial Myo-II enriched pool, with the similar period and 1 min-delay (Fig. 2g–l and Supplementary Movie 5). Taken together, our results demonstrate that ROCK, Myo-II, and local MLCP display similar, but temporally shifted oscillatory patterns at the basal medial cortex (Fig. 2j, l), while membrane Rho1 level and activity remain constant (Fig. 2k). This indicates that, similar to observations in the *Drosophila* embryo[20], Myo-II basal oscillations are not dictated by an upstream Rho1 oscillation, thus leading us to ask whether oscillations could be intrinsically encoded in the signaling network downstream of Rho1.

### Oscillation is not dependent on actomyosin contraction force.

A key question is whether such oscillations derive from the mechanical or chemical properties of actomyosin networks. Using Y27632 to inhibit the ROCK kinase activity and thus Myo-II contraction[20] cannot answer this question precisely, as it is expected to affect both mechanics, via changing the contractility of Myo-II, and the chemical network by blocking Rock kinase

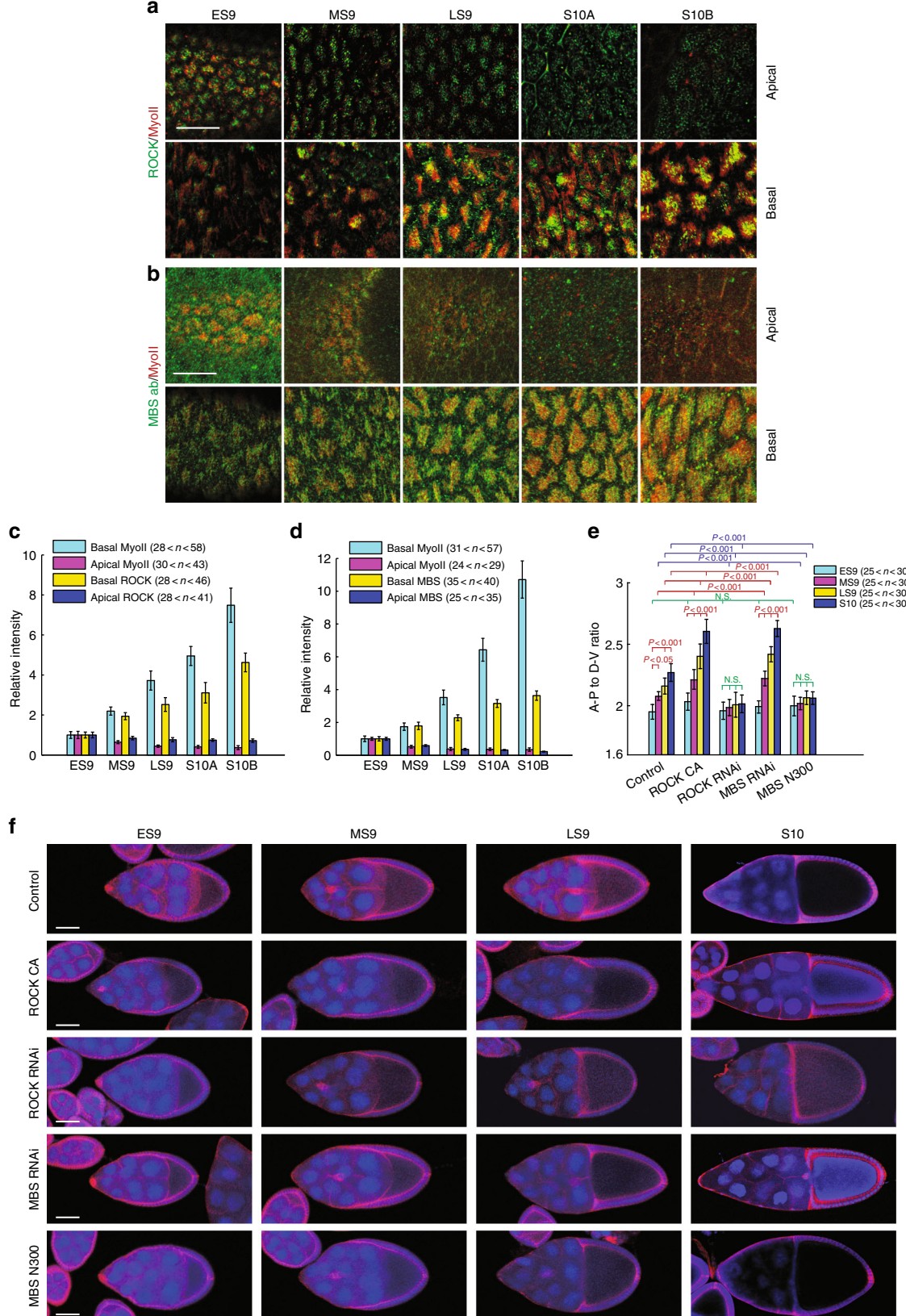

activity. Instead, to test whether oscillations derive from the mechanical properties of the cytoskeleton, as hypothesized in previous studies[10,11], we mosaically expressed a dominant negative form of MHC[28] in follicle cells. In control cells, Myo-II

oscillation is followed by periodic change of basal membrane area reduction (by around a minute, Fig. 3a, b, e), as expected from the contractile properties of actomyosin to reduce basal membrane area[9]. Considering its role in blocking the Myo-II loading to F-

actin[28], overexpression of dominant negative MHC inhibits the actomyosin contractility, and accordingly, no prominent change of basal area reduction can be observed in this condition (Fig. 3c, d). Strikingly however, this inhibition of actomyosin contractility did not affect basal Myo-II oscillation; strong Myo-II oscillations are still present, with periods and intensities undistinguishable from those of control cells (Fig. 3c–g). These results confirmed that actomyosin contractility does not control basal Myo-II oscillation, and thus rather suggest that, in this system, membrane area changes are a passive consequence of Myo-II oscillation, rather than an active feedback, in contrast to a previous model[10]. This led us to investigate whether basal Myo-II oscillation could instead arise from a biochemical oscillator.

**Oscillations initiate from a ROCK flow beginning at junction.** First question that arises is the nature of the Myo-II activator initiating a cycle of oscillation. As Rho1, the better characterized upstream activator of Myo-II, is restricted to the membrane, we asked whether and how Rho1 could provide cues to the medial-restricted Myo-II pool. Interestingly, we observed that ROCK oscillation starts at the basal junctional membrane, where Rho1 resides, and then propagates inward (Fig. 4a, b and Supplementary Movie 6). To quantify this, we applied a "mean field spatial analysis" method named as "optical flow" analysis after denoising of raw images (see Methods section), and detected strong flow movements for both ROCK and Myo-II signals in basal domain of oscillating follicle cells (Fig. 4c). ROCK flow movements occur earlier than those of Myo-II around 1 min (Fig. 4d, e), which is similar to the intensity temporal delay between these two proteins at the medial basal pool (Fig. 2j, l). Interestingly, this is in stark contrast to intercalating ectoderm cells, characterized by the inverse patterns between flow movement and signal intensity[20].

This suggests that the initiation of intensity increase and movement could be dependent on the ROCK interaction with membrane active Rho1, which unfolds and activates ROCK[29]. To test this hypothesis, we used the chemical inhibitor Rhosin to block Rho1 activity in order to determine the effect on both intensity change and flow movement of ROCK signals. Compared with the control, treatment of Rhosin strongly results in the loss of ROCK distribution at basal junction (Fig. 4f–h). Consistently, follicle cells with the Rho1 inhibition have much weaker intensity increase and flow movement of ROCK signals (Fig. 4g, i). Thus, the interaction of ROCK with membrane active Rho1 appears to initiate both signal increase and flow movement. However, as Rho1 activity and concentration do not oscillate, such an interaction alone would simply produce a uniform concentration of ROCK. From the theoretical perspective, activator/inhibitor type dynamics have the potential to create such self-organized oscillatory patterns, even in the absence of possible upstream regulators (other than Rho1) that could be oscillatory[30].

**Positive and negative regulators of basal ROCK signals.** Therefore, we checked how activating or inhibiting the members of the Rho1 to MLCP signaling cascade affected the levels and

dynamics of wild type (WT) ROCK. As expected, active and dominant negative forms of Rho1 respectively increase and decrease ROCK concentration in the basal medial pool (Fig. 5a and Supplementary Fig. 3a). It is believed that ROCK acts as an inhibitor of MBS, thus relieving the inhibitory effect of MLCP on Myo-II[31]. Surprisingly, inhibiting (resp. constitutively activating) MBS increased (resp. decreased) basal medial ROCK concentration 2-fold, arguing for the converse negative feedback, either direct or indirect, of MBS on ROCK (Fig. 5a and Supplementary Fig. 3a). Moreover, overexpression of an active form of ROCK strongly enhanced the accumulation of WT ROCK in the basal medial pool, indicating that open and active ROCK might be able to amplify the accumulation and flow movement of WT ROCK molecules. To validate these findings, we repeated the experiments while measuring ROCK levels via antibody staining instead, and found the same results (Supplementary Fig. 4).

Next, we hypothesized that this ROCK amplification could be either via direct control of ROCK accumulation, i.e., via an oligomerization mechanism with auto or trans-phosphorylation between different ROCK molecules[29], or via the indirect control of Myo-II phosphorylation, in which the formation of Myo-II mini-filaments would serve as a scaffold for additional ROCK accumulation. To test the latter hypothesis, we investigated the effect of Myo-II phosphorylation-mimic mutant (Myo-II EE) on ROCK levels, and found on the contrary an inhibitory effect compared to control and unphosphorylatable mutant (Myo-II WT form and Myo-II AA form, in Fig. 5a and Supplementary Figs. 3a–f, 4e, f). A recent study demonstrated that Myo-II EE had much weaker motor activity and thus less contractile property[32]. However, in the presence of WT genetic background, over-expression of Myo-II EE in follicle cells strongly enhanced the formation of stress fibers and also the elongation of egg chambers, while overexpression of Myo-II AA significantly reduced both (Supplementary Fig. 3g–j). The different effects of Myo-II EE vs. AA forms on ROCK levels, together with on stress fibers and actomyosin contractility, contradicted the hypothesis of the Myo-II mini-filament–mediated ROCK signal amplification. Interestingly, although the aforementioned overexpression of dominant negative MHC inhibited the actomyosin contractility (Fig. 3c, d), we found that this did not affect the basal medial ROCK levels (Supplementary Fig. 4e, f). This further indicates that actomyosin contractility is not necessary for ROCK signal self-amplification. These data thus hint to the fact that ROCK kinase activity could self-amplify ROCK signals via its oligomerization mechanism.

Altogether, our findings demonstrate that ROCK can both display self-amplification, and be repressed by MBS.

**Modeling of oscillatory characteristics and phase shift.** Given the uncovered regulatory interactions, we next asked whether these would be sufficient, from a theoretical perspective, to give rise to self-organized Myo-II oscillations. We therefore mathematically modeled the evolution in time of ROCK, Myo-II, and MBS/flw concentrations (resp. denoted $R$, $m$, and $M$). Before asking about spatial flow patterns, we first made the simplifying

**Fig. 1** ROCK, Myo-II, and MBS signals have an apical to basal distribution switch during the periods of basal Myo-II oscillation and control morphogenetic tissue elongation. **a**, **b** Apical and basal views of follicle cells from egg chambers at early stage 9, middle stage 9, late stage 9, stage 10A, and stage 10B, marked by the presence of MyoII-mCherry (red) with ROCK-GFP (green) (**a**), and MyoII-mCherry (red) with MBS-antibody staining (green) (**b**), respectively. Both scale bars are 30 μm. **c**, **d** Quantifications of relative apical and basal ROCK/Myo-II intensities (**c**) and MBS/Myo-II intensities (**d**) at different stages. **e** Quantification of the A–P to D–V length ratio in early stage 9, middle stage 9, late stage 9, and stage 10 egg chambers expressing the indicated transgenes. **f** Morphology of early stage 9, middle stage 9, late stage 9, and stage 10 egg chambers expressing the indicated transgenes, staining by Armadillo and DAPI, 4′,6-diamidino-2-phenylindole. All scale bars are 50 μm. $n$ is the number of samples (cells in **c** and **d** from four independent egg chambers) analyzed. Error bars indicate ±s.d. NS means not significant, $p < 0.001$ means significant difference by Student's $t$-test

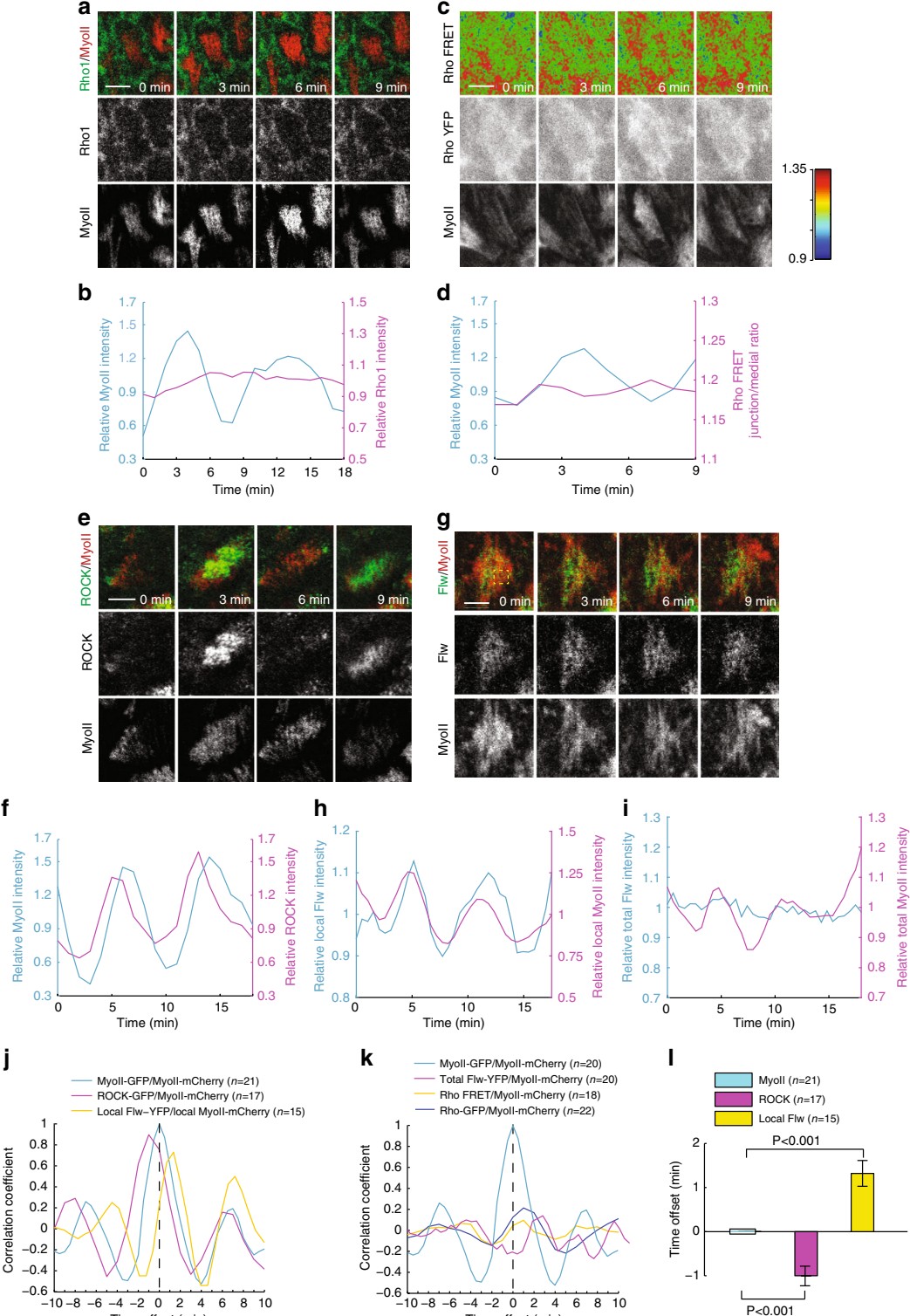

**Fig. 2** Myo-II pulses with its regulators ROCK and Flw at the medial basal cortex but not with basal junction Rho1. **a**, **c**, **e**, **g** Time-lapse series of the representative oscillating follicle cells, labeled with Rho1-GFP and MyoII-mCherry (**a**), Rho FRET activity and MyoII-mCherry (**c**), ROCK-GFP and MyoII-mCherry (**e**), Flw-YFP and MyoII-mCherry (**g**), respectively. In **c**, Top panel is the processed Rho FRET signal; middle panel is YFP channel only; bottom panel is Myo-II signal. All scale bars are 5 μm. **b**, **d**, **f**, **h**, **i** Quantifications of the dynamic change of Rho1-GFP and MyoII-mCherry intensities (**b**), Rho FRET activity and MyoII-mCherry intensity (**d**), ROCK-GFP and MyoII-mCherry intensities (**f**), Flw-YFP and MyoII-mCherry intensities (**i**) in one oscillating cell, and Flw-YFP and MyoII-mCherry (**h**) in a local basal medial region of one oscillating cell (marked by dotted square in **g**), respectively. Intensity of each channel is normalized to its mean in **b**, **d**, **f**, **h**, **i**, except that Rho FRET activity is original value in **d**. **j**, **k** Average temporal cross-correlation of MyoII-mCherry with MyoII-GFP (cyan), ROCK-GFP (magenta) and local Flw-YFP (yellow) (**j**), MyoII-mCherry with MyoII-GFP (cyan), total Flw-YFP (magenta), Rho FRET activity (yellow) and Rho1-GFP (blue) (**k**). **l** Time offset calculated from the cross-correlation analysis. $n$ is the number of samples analyzed. All error bars indicate ±s.d. $p < 0.001$ means significant difference by Student's $t$-test

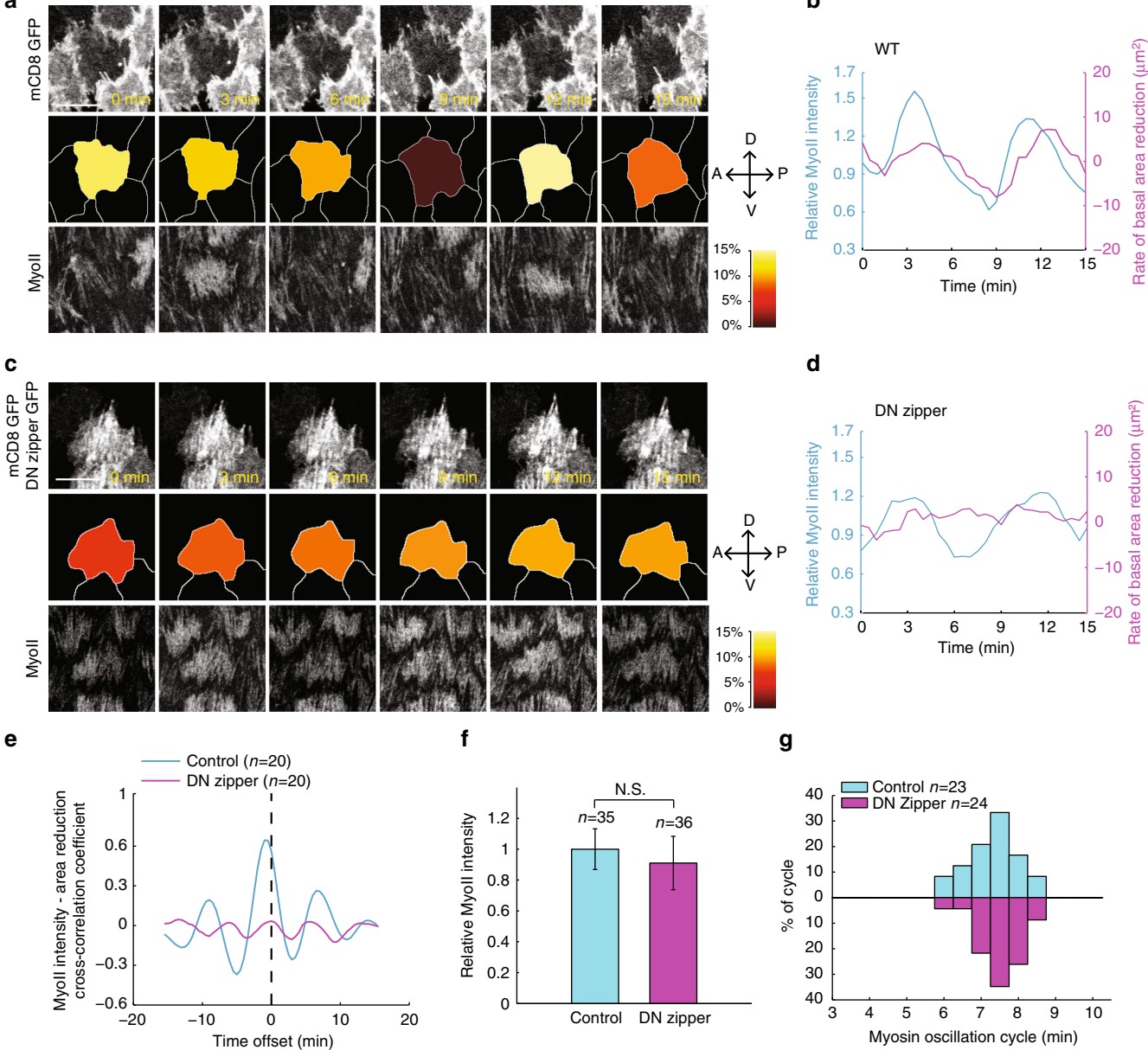

**Fig. 3** Myo-II undergoes periodic accumulation and disassembly no matter whether Myo-II is able to be loaded on F-actin or not. **a**, **c** Time-lapse series of the representative wild type (**a**) and dominant negative MHC-expressing (**c**) follicle cells, labeled with mCD8-GFP and MyoII-mCherry. Both scale bars are 10 μm. The digitized cell contour is color-coded on the basis of the percentage decrease in surface area relative to the maximum area captured during imaging, as indicated in the heat map. **b**, **d** Quantifications of the dynamic change of relative Myo-II intensity and rate of basal area reduction from the same wild type (**b**) and dominant negative MHC-expressing (**d**) follicle cells. **e** Myo-II intensity-area reduction cross-correlation coefficient in the wild type vs. dominant negative MHC-expressing follicle cells. **f**, **g** Quantifications of relative Myo-II intensities (**f**) and oscillating cycle time period (**g**) in the indicated transgene-expressing GFP-positive cells compared with wild-type cells in the same tissue sample. *n* is the number of samples analyzed. All error bars indicate ±s.d. NS means no significant difference by Student's *t*-test

assumption of uniform basal concentrations which in non-dimensional form reads

$$\frac{\mathrm{d}R}{\mathrm{d}t} = \frac{R}{\tau_R}(1 - M)$$

$$\frac{\mathrm{d}m}{\mathrm{d}t} = -\frac{m}{\tau_m}(1 - AR + BM) + \frac{J}{\tau_m}$$

$$\frac{\mathrm{d}M}{\mathrm{d}t} = \frac{(m - M)}{\tau_M}$$

where the model has been reduced to three equations with five

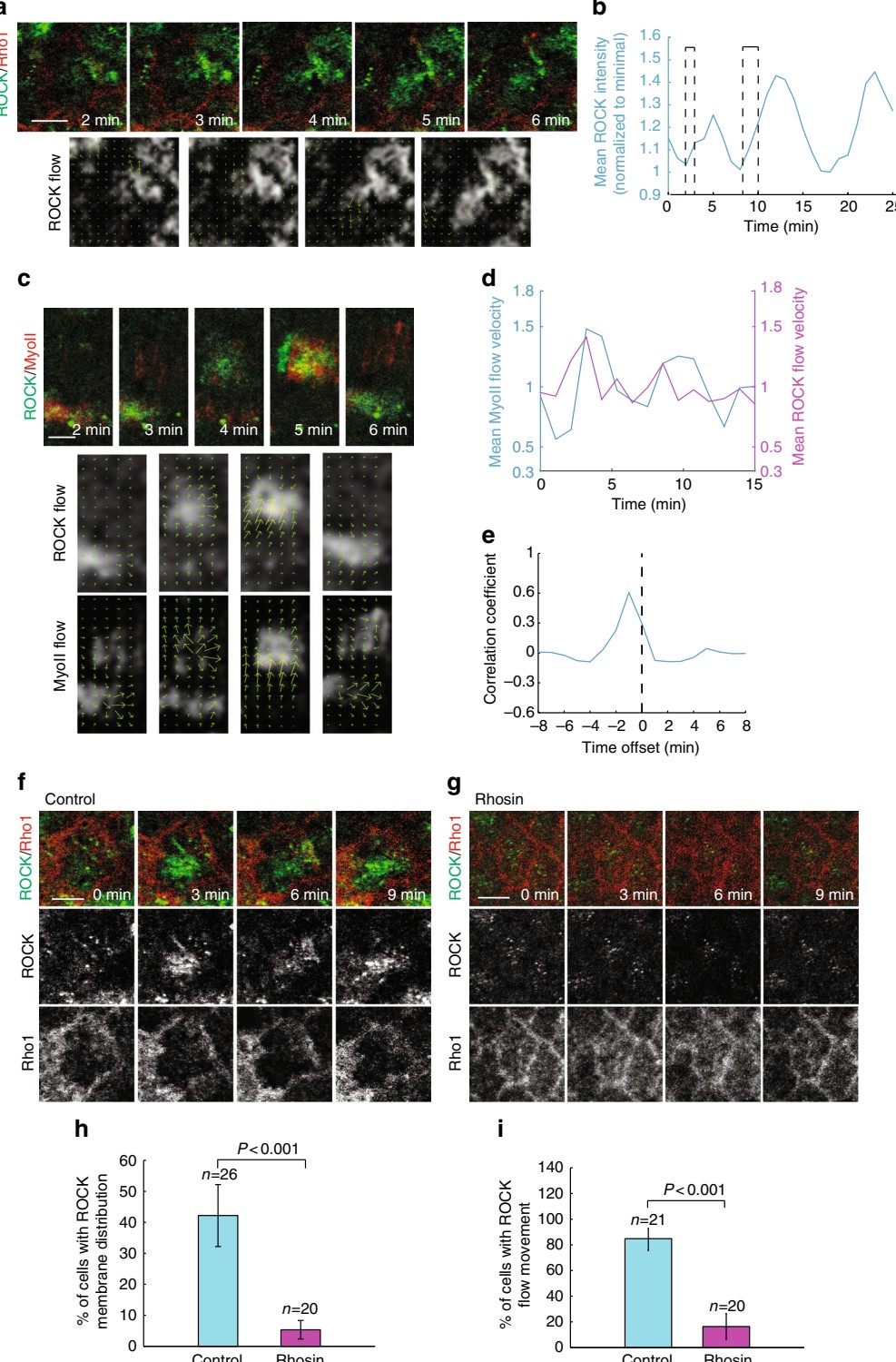

**Fig. 4** Myo-II pulsatility follows a flow movement of ROCK initiated at basal junction. **a** Time-lapse series of one representative follicle cell labeled with ROCK-GFP and Rho1-mCherry, followed by the time-lapse series analysis of flow velocity of ROCK signals. **b** Quantification of the dynamic change of mean ROCK intensity, which is normalized to its minimal value. The rectangle region is the period when ROCK is in close proximity to Rho1 at and near the basal junction. **c** Dynamics of ROCK and Myo-II signals by the analysis of "Optical flow" method. Upper panels show the time-lapse series of ROCK-GFP and MyoII-mCherry signals in one representative oscillatory follicle cell basal domain. Lower panels show the time-lapse series analysis of flow velocity of ROCK and Myo-II signals. **d** Quantification of the dynamic change of mean Myo-II and ROCK flow velocities in one oscillating follicle cell. Velocity of each channel is normalized to its mean. **e** Average temporal cross-correlation of Myo-II velocity with ROCK velocity. **f**, **g** Time-lapse series of the representative oscillating follicle cell labeled with ROCK-GFP and Rho1-mCherry, in either control (**f**) or Rhosin treatment condition (**g**). All scale bars are 5 μm. **h** Quantification of the percentage of cells with ROCK membrane distribution in control vs. Rhosin treatment conditions. **i** Quantification of the percentage of cells with ROCK flow movement in control vs. Rhosin treatment conditions. $n$ is the number of samples analyzed. All error bars indicate ±s.d. $p < 0.001$ means significant difference by Student's $t$-test

free parameters: the three characteristic time scales of ROCK, Myo-II, and MBS/flw recruitment ($\tau_R$, $\tau_m$, $\tau_M$), and two interactions coefficients $A$ and B, respectively describing the effect of ROCK and MBS/flw on Myo-II. As detailed in Supplementary Note 2, we found that the aforementioned combination of ROCK self-activation and ROCK inhibition by MBS/flw was sufficient to generically generate biochemical oscillations. Moreover, the model qualitatively reproduced well the time-delay between ROCK, Myo-II, and MBS/Flw successive oscillations (Fig. 5b, d and Supplementary Note 2). To be able to make more quantitative predictions, we wished to contain the parameter set, and quantified, in particular, the recovery times of all three components. We thus performed Fluorescence Recovery after Photobleaching (FRAP) experiments, with confocal variable volumes (see Methods section and Supplementary Note 1) to calculate diffusion coefficients and characteristic recovery times from each protein of interest. A classical diffusion fitting law showed different recoveries times $\tau_i$ for Myo-II (half-life time of recovery: 60 ± 30 s), ROCK (15 ± 7 s), and Flw (38 ± 15 s) (Fig. 5c and Supplementary Fig. 5). We thus used these as inputs for our model. Importantly, a theoretical prediction is that the delays between ROCK and Myo-II (resp. Myo-II and MBS/Flw) should be closely related to the characteristic recovery time of Myo-II (resp. MBS/Flw), see Supplementary Note 2 and Supplementary Fig. 6a–e, l for a phase diagram. Generically, a theoretical sensitivity analysis showed that the period of the oscillation $T_p$, as well as the delays between ROCK, Myo-II ($t_1$), and MBS/Flw ($t_2$) were robust with respect to the remaining two parameters, and fitted well the data (Fig. 5d and Supplementary Note 2 for details, $T_p \approx 7$ min, $t_1 \approx 50$ s, $t_2 \approx 40$ s), so that the model reproduces accurately the observed oscillations and time sequence between the three components.

**Quantitative prediction of perturbation on the oscillator**. Next, we further tested our biochemical oscillator model. In particular, we sought to predict quantitatively the mutant experiments via in silico considerations. We first tested the effect of modifying the activity of Rho1, and found that above a certain value, it desensitizes the system, destroying oscillations, reminiscent of the effect of the external potential in the FitzHugh-Nagumo model of spiking neurons[33] (Supplementary Fig. 6h and Supplementary Note 2). Importantly, experimental activation and inhibition of Rho1 strongly increase and decrease basal Myo-II intensity respectively, both with Myo-II oscillation becoming undetectable (Fig. 6a–h). We then investigated theoretically the effect of modifying the activity of ROCK and MBS/Flw, keeping all other parameters constant (Supplementary Note 2 for details), and found for both, a robust positive correlation between the resulting period and amplitude of Myo-II oscillations (Fig. 5e–g and Supplementary Fig. 6a–e). Importantly, we tested experimentally this unexpected prediction, and found that activation and inhibition of ROCK respectively increase and decrease both Myo-II oscillatory amplitude and cycle time period (Fig. 6i–p). Strikingly, activation and inhibition of MBS respectively decrease and increase both Myo-II oscillatory amplitude and cycle time (Fig. 6q–x). This confirmed experimentally the model prediction of a robust positive correlation between the period of the oscillation and its amplitude (Fig. 5g). One should note that in the model, the amplitude of oscillation is more sensitive to change in ROCK/MBS activity than the experimental counter-part, which could be due to saturation effects in the kinetic coefficients entering in our simplified model, and/or additional secondary feedbacks.

**Modeling of spatio-temporal oscillations**. In a second model step, we wondered whether we could reproduce the observed diffusive flow of ROCK signals in the model scheme. For this, we introduced the spatial counter-part to the above model, assuming the exact same reaction kinetics, but additionally allowing ROCK, Myo-II, and Flw to diffuse spatially on the basal surface, as observed from the aforementioned FRAP experiments (Supplementary Fig. 5; see Methods section). Interestingly, we found that Myo-II recovery displays minimal 2D diffusion in the basal plane, and is thus dictated by turnover from the cytoplasmic pool (Supplementary Fig. 5). For ROCK and Flw, the recovery is with turnover from the cytoplasmic pool, but also a significant 2D diffusion along the basal area (Supplementary Fig. 5), which provides a potential explanation for the centripetal flows of ROCK and Flw observed during oscillations. Importantly, as Rho1 is only present at the edge of cell, numerical simulations of the model predict that ROCK is first activated at the boundary, and then propagates as a kinematic wave towards the cell basal center at a constant velocity, mirroring the data, before being "caught up" by MBS/Flw at the edge (Fig. 5h–j for a 1D numerical integration and Supplementary Fig. 6i–k for a 2D numerical integration).

**Confirmation of the biochemical oscillator by optogenetics**. To further and directly probe the spatio-temporal dynamics of the biochemical oscillator, we introduced an optogenetic method named as GFP-LARIAT (Supplementary Fig. 7a), which we recently established in *Drosophila* in vivo to create a light-inducible GFP trap and thus disturb the localization and function of GFP-tagged proteins[23,34]. Here, we used GFP-LARIAT to induce the clustering trap of ROCK-GFP, Flw-YFP, or MBS-GFP in order to assess the effect on Myo-II. Firstly, we confirmed that the clustering trap of ROCK-GFP or Flw-YFP/MBS-GFP by light-induced GFP-LARIAT reduces or enhances the phosphorylation of MRLC, respectively, in follicle cells, compared with control cells (Fig. 7a–d and Supplementary Fig. 8a–c). Clustering trap of ROCK-GFP gradually and strongly reduces pulsatile Myo-II signals, while clustering trap of Flw-YFP or MBS-GFP gradually and significantly enhances Myo-II oscillations, compared with control conditions (Fig. 7e–k, Supplementary Figs. 7b–g, 8d–i, and Supplementary Movies 7–12). Before light illumination, both ROCK and Flw/MBS are highly co-localized with Myo-II. With the clustering progression, both ROCK and Flw/MBS tend to separate from Myo-II, indicating the functional loss of GFP-tagged proteins in control of downstream signals, compared with control (Supplementary Figs. 7h, i, 8j). Therefore, our optogenetic results fully confirmed in a more direct manner the main conclusion acquired by genetics and modeling.

**Discussion**

Here, we have demonstrated that a biochemical oscillator involving ROCK and MLCP, two key regulators of Myo-II, controls basal Myo-II oscillation, which is important in normal *Drosophila* egg chamber elongation. ROCK functionally links its upstream control (active Rho1 at and near basal junctional membrane) with its downstream substrates (Myo-II at basal medial region). This link is dependent on a directional flow movement of ROCK from junctional membrane to medial cortical region. As ROCK shuttles medially, the signals of ROCK get self-amplified by its kinase activity-dependent accumulation, which leads to phosphorylation and mini-filament accumulation of Myo-II. This in turn governs the redistribution of MLCP signals toward the regions of Myo-II

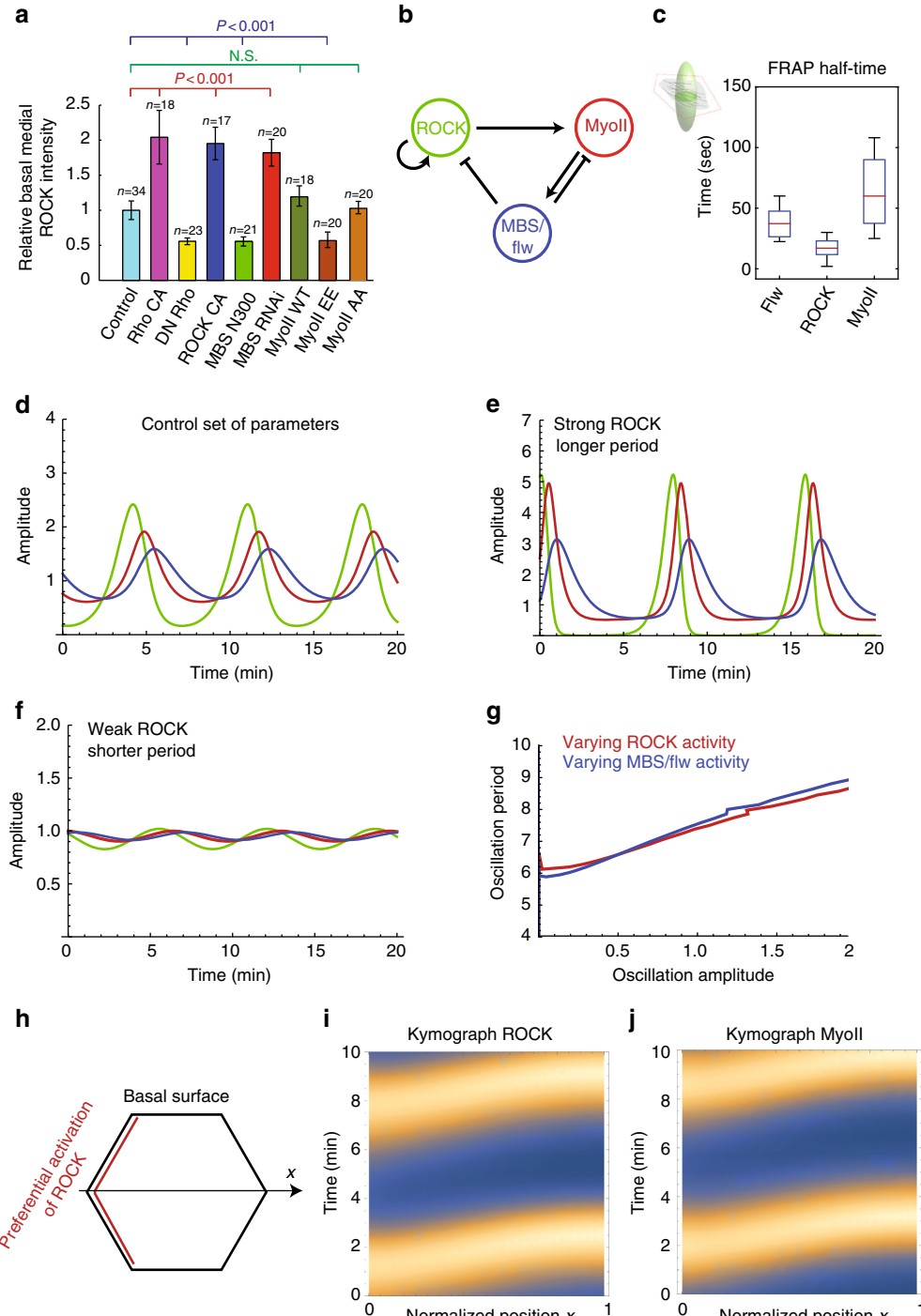

**Fig. 5** In silico predictions of basal Myo-II oscillations and flow movements. **a** Quantification of relative ROCK-GFP intensity in the basal medial pool from the indicated transgene-expressing mCD8RFP-positive cells compared with wild-type cells in the same sample. The representative images have been shown in Supplementary Fig. 3a, from which ROCK-GFP signals at the basal medial region have been analyzed. *n* is the number of samples analyzed. All error bars indicate ±s.d. NS means no significant difference, while *p* < 0.001 means significant difference by Student's *t*-test. **b** Summary schematics of the inferred interaction diagram between ROCK, Myo-II, and MBS/Flw, showing both positive and negative interactions. **c** The half-life recovery time of Flw-YFP, ROCK-GFP, and MyoII-GFP after photobleaching. **d** In silico predictions on the dynamics of the ROCK/Myo-II/Flw system, based on the interactions of **b** and the measured FRAP recovery kinetics in **c**, matching the experimentally observed oscillations. **e**, **f** Increasing (resp. decreasing) the activity of ROCK theoretically increases (resp. decreases) both the period and the amplitude of the oscillation (the control parameters were A = 0.8 and B = 0.5). **g** Predicted correlation between oscillation amplitude and period upon manipulation of either ROCK (red) or MBS/Flw (blue) activity. **h–j** Schematics (**h**) of adding a spatial component to the model; we assume that ROCK is preferentially activated on the junctional membrane due to Rho1 localization, and corresponding simulated kymographs for ROCK (**i**) and Myo-II (**j**) showing that oscillations still occur, but start at the junctional membrane, resulting in a diffusive flow from membrane to center

accumulation, in order to inactivate ROCK and Myo-II, and restart a cycle (Supplementary Fig. 9).

This biochemical oscillator controlling basal Myo-II contractility shares some similarities with the recently reported oscillatory signaling cascade that regulates apical Myo-II contraction in the intercalating ectoderm cells[20]; both oscillatory signaling cascades contain ROCK, Myo-II, and MBS, and there is a significant temporal delay between Myo-II and MBS signals,

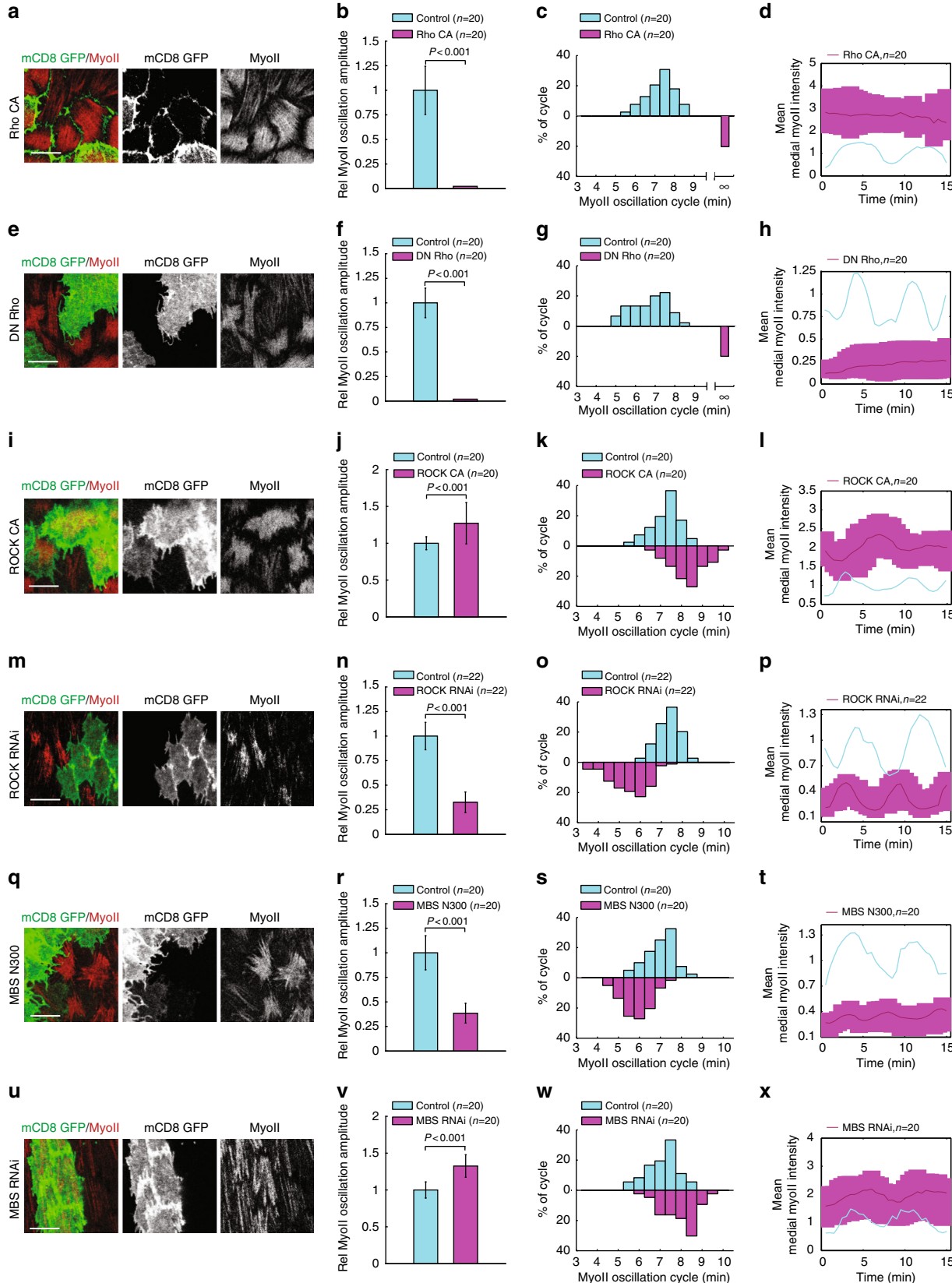

which not only allows the formation of biochemical oscillation but also shut off all biochemical signals. However, these two oscillatory systems are quite different in several aspects. Firstly, Rho1 seems to be only transiently in contact with ROCK but not to Myo-II or MBS in basal Myo-II oscillation, while Rho1 moves together with ROCK and actomyosin networks in apical Myo-II oscillation. This difference might stem from the subcellular localization variation of the Rho1–Myo-II signal cascade. Secondly, Myo-II signals have prominent spatio-temporal delay from ROCK signals during basal Myo-II oscillation, while ROCK and Myo-II signals are synchronized in ectoderm cells. Thirdly, we propose that ROCK/Myo-II flows in our system are diffusive and biochemical in nature, as the inhibition of actomyosin contractility (via the expression of the dominant negative form of MHC) did not affect ROCK levels or dynamics, providing a complementary mechanism to the force-driven physical advection of the actomyosin contraction networks. A possible explanation for the large difference in time cycle between these two Myo-II oscillations would be that the basal condensed stress fibers might block the physical advection of biochemical molecules and thus slow down the speed of flow movement and oscillating cycle period. It was striking in this regard to observe from FRAP experiments that Myo-II does not significantly diffuse laterally, with most of its dynamics associated with exchange kinetics with the cytoplasmic pool.

In the oscillatory ectoderm cells, apical E-cadherin difference along DV and AP axis membrane plays a major role in controlling the specific direction of Myo-II flow movement from the junctional membrane to medial apical cortex[8,35]. However, basal E-cadherin levels are quite equal along DV vs. AP membrane[23], indicating that basal Myo-II flow movement is not due to the E-cadherin tension variance. We also excluded the possibility that the contraction of medial Myo-II motors triggers the flow of ROCK signals. Our studies indicate that signaling spatial difference is critically important in the formation of flow movements and signal oscillations (Figs. 2 and 5h–j). How exactly ROCK signals get self-amplified by its kinase activity, and then disassembled by MBS during flow movements, would be an important next step to further dissect molecularly this oscillation. In particular, the amplification and disassembly of ROCK signals might be via the direct interaction of ROCK/ROCK or ROCK/MBS, or via another intermediate factor which is linked with and possibly controlled by both ROCK and MBS.

Interestingly, although biomechanical effects, including actomyosin contractility, are dispensable for basal Myo-II oscillation, they still have key roles for morphogenesis. It is noteworthy that the occurrence of this biochemical oscillator is concomitant to strong tissue shape change. Indeed, any misregulation of the factors in this oscillator strongly changes the underlying tissue shape. Thus, it remains unclear if the oscillator has a defined function for morphogenesis, or if it is a necessary consequence of increased contractility, due to the structure of the ROCK/Myo-II/MBS regulatory network. Interestingly, however, different from the defective phenotype induced by exacerbated apical Myo-II contractility, enhanced basal Myo-II contractility results in the stronger phenotype of egg chamber elongation[9,23]. It indicates that basal pulsatile actomyosin contractility might function to create a gradual, but not acute, tissue elongation during morphogenesis so that a functional organ is precisely developed.

## Methods

**_Drosophila_ stocks and genetics**. The following fly stocks were used: Sqh::RLCmyosinII–mCherry and Sqh::RLCmyosinII-GFP[6] (from Eric E. Wieschaus), DN Zipper (from Daniel Kiehart), Ubi::ROCK–GFP[36] (from Yohanns Bellaiche), UAS-ROCK[RNAi], UAS-MBS[RNAi] (from Vienna _Drosophila_ RNAi center), UAS-Sqh WT, UAS-Sqh EE, UAS-Rho CA, UAS-ROCK CA, UAS-MBS N300, Rho1-GFP, Rho-mCherry, UAS-Rho DN, ROK[1]/FM7 (from Bloomington _Drosophila_ stock center), hsGal4/CyO, MKRS/TM6B was used to express UAS lines in follicle cells in the experiments of Rho FRET dynamics. For ROCK/myo-II and ROCK/Rho1 dynamics, Ubi::ROCK–GFP on the third chromosome were combined with ROK[1]/FM7 mutant flies in order to prevent the side effect of ectopic ROCK overexpression. Clones were generated using FLP-OUT technique by crossing UAS transgenic flies with: (1) P[hsp70-flp]; Sqh::Sqh–mCherry; UAS-mCD8GFP, AyGal4; (2) P[hsp70-flp];+/+; UAS-mCD8GFP, AyGal4; (3) P[hsp70-flp];+/+; Ay(CD2)Gal4. To detect ROCK signal in mosaic clones, Ubi::ROCK-GFP on the X chromosome were needed to be combined with UAS transgenic flies first, and then clones were generated using FLP-OUT technique by doing the second cross with P[hsp70-flp];+/+; UAS-mCD8RFP, AyGal4. All stocks and crosses were maintained at room temperature. For signal analysis in mosaic clones, hsFLPase was induced for 1 h at 37 °C twice with a 5 h interval, then flies were kept at 18 °C for 2 days and then fattened at 25 °C for overnight before dissection. For the analysis of tissue elongation, P[hsp70-flp];+/+; Ay(CD2)Gal4 was used for cross and later heat shock treatment, which can induce more than 90% clone cells in egg chamber. hsFLPase of this mosaic system was induced for 1 h at 37 °C once, then the flies were kept at 18 °C for 1–2 days and then fattened at 25 °C overnight before dissection. For the experiments of Rho FRET dynamics, hsGal4 flies were incubated at 37 °C for 1 h and the flies were kept at 25 °C for 5–6 h before dissection. For LARIAT experiments, hsGal4 was used to induce LARIAT expression in ovary follicle cells, and all steps were carried on in dark condition, including cross, maintenance, and heat shock. Flies were treated with heat shock at 37 °C for 1 h, and then incubated 2–3 h at 25 °C before dissection. _Drosophila_ ovaries were dissected and egg chambers were mounted under red light before blue light illumination.

**DNA constructs and transgenic fly generation**. Full length MBS cDNA was obtained from Genebank and amplified by primers listed below. The plasmids of Ubi-MBS-GFP and Ubi-MBS-RFP were generated by in-fusion cloning (Clontech). The flies were generated by Bestgene Inc. using w1118 fly.

**PCR to amplify MBS**. Forward 5′-GGTGGAGGTGGTGGTATGTCCTCGCTGGACGC-3′
 Reverse 5′- AATTGGGGTACGTCTAGACCAACTGGTAATGGTAGCGAC-3′

**PCR to amplify GFP**. Forward 5′-CCGGGCTGCAGGAATTCATGGTGAGCAAGGGCGAG-3′
 Reverse 5′- ACCACCACCTCCACCCTTGTACAGCTCGTCCATGC-3′

---

**Fig. 6** Experimental perturbation confirmation of in silico prediction on basal Myo-II oscillations. **a**, **e**, **i**, **m**, **q**, **u** Basal views of follicle cell clones expressing the indicated transgenes, marked by coexpression of mCD8GFP. Myo-II signals are monitored by MyoII-mCherry. All scale bars are 10 μm. **b**, **f**, **j**, **n**, **r**, **v** Quantifications of relative Myo-II oscillation amplitude in the indicated transgene-expressing GFP-positive cells compared with wild-type cells in the same sample. **c**, **g**, **k**, **o**, **s**, **w** Quantifications of relative Myo-II oscillating cycle time period in the indicated transgene-expressing GFP-positive cells compared with wild-type cells in the same sample. For both Rho CA-expressing and DN Rho-expressing cells, basal Myo-II oscillations are almost undetectable. Thus, relative Myo-II oscillation amplitude is zero in these two conditions. Symbol ∞ means that oscillating cycle time period is not prominent, also in these two conditions. The relative Myo-II average intensity from these analyzed basal Myo-II oscillations are 1 ± 0.23 vs. 2.98 ± 0.79 in control vs. Rho CA cells, 1 ± 0.20 vs. 0.19 ± 0.09 in control vs. DN Rho cells, 1 ± 0.27 vs. 2.17 ± 0.45 in control vs. ROCK CA cells, 1 ± 0.16 vs. 0.27 ± 0.11 in control vs. ROCK RNAi cells, 1 ± 0.23 vs. 1.95 ± 0.52 in control vs. MBS RNAi cells, and 1 ± 0.21 vs. 0.36 ± 0.12 in control vs. MBS N300 cells, respectively. **d**, **h**, **l**, **p**, **t**, **x** Quantifications of the dynamic change of medial basal mean Myo-II intensity in the indicated transgene-expressing cells compared with one wild-type cell in the same sample. Each pink curve is the dynamic change from the average of Myo-II intensities in different cells with a range marked by pink color, and blue curve is the dynamic change of Myo-II intensity in one representative wild-type cell. n is the number of samples analyzed. All error bars indicate ±s.d. $p < 0.001$ means significant difference by Student's $t$-test

**PCR to amplify mCherry.** Forward 5′-
CCGGGCTGCAGGAATTCATGGTGAGCAAGGGCGAGG-3′
   Reverse 5′- ACCACCACCTCCACCCTTGTACAGCTCGTCCATGC-3′

**Production of ROCK antisera.** To produce anti-ROCK antibody, a peptide corresponding to the C-terminus of *Drosophila* Rok-PA, amino acids 1350–1370 (CYNNNSTDGSKISPSQSTRSSY), was synthesized by YenZym Antibodies, LLC (South San Francisco, CA). The N-terminal C was added to couple the peptide to the KLH carrier for immunization. Two rabbits were immunized by YenZym,

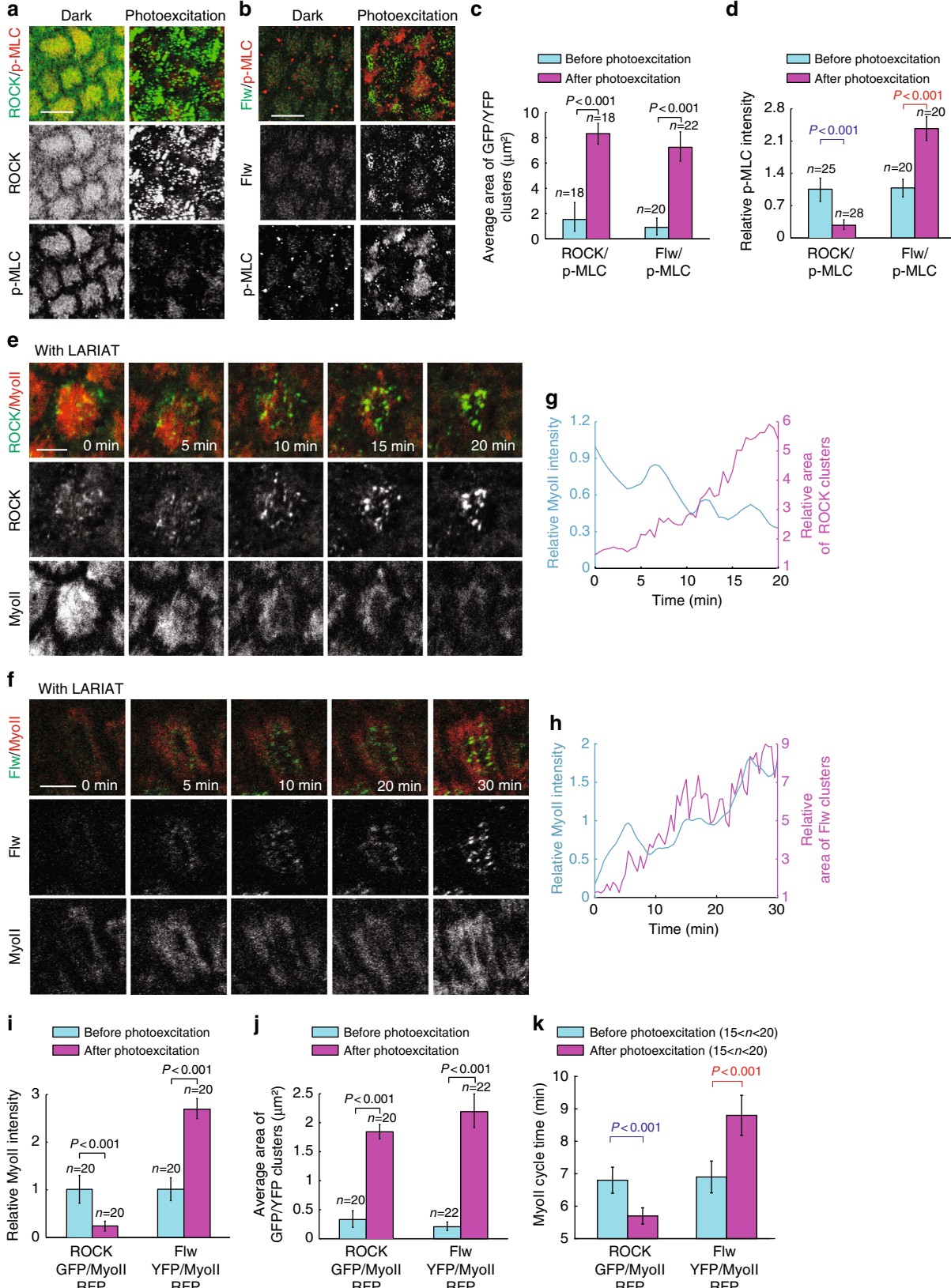

YZ3971, and YZ3972. YZ3972 had the best response by ELISA and showed specific signal on fixed tissue at 1:2000 and recognized a predicted 160 kDa band.

**Dissection and mounting of *Drosophila* egg chamber**. One to three-day-old females were fattened on yeast with males for 1–2 days before dissection. *Drosophila* egg chambers were dissected and mounted in live imaging medium (Invitrogen Schneider's insect medium with 20% FBS and with a final PH adjusted to 6.9), by using a similar version of the protocol described in ref. [37]. Different from normal mounting condition, the egg chambers were slightly compressed to overcome the endogenous curvature. In this condition, basal oscillation pattern, intensity, and period were similar to those in the condition without compression.

**Imaging and FRET**. Time-lapse imaging was performed with a Zeiss LSM710 or Leica SP8 confocal microscope with a ×40, numerical aperture 1.3 inverted oil lens, a 488-nm argon laser, and a 568-nm green HeNe laser. The basal focal plane, which is about 1 µm beneath the basal surface, was selected during live imaging to maximize the basal Myo-II intensity. For the dynamics of ROCK–GFP, MBS–GFP, MBS-RFP, or Flw-YFP, the similar basal focal plane was selected to maximize the basal signal intensity, as basal MyoII dynamic imaging did. The same microscope setup was used when comparing intensity between different samples. To view the signals at different focal planes, images were taken at different Z-stack layers from the basal surface to the apical side.

FRET images of live cultured egg chambers were acquired with a Zeiss LSM710 microscope, by using a similar version of the protocol described in ref. [38]. A 458 nm laser was used to excite the sample. CFP and YFP emission signals were collected through channel I (470–510 nm) and channel II (525–600 nm), respectively. To capture single, high-resolution, and stationary images, a ×40/1.3 oil inverted objective was used. CFP and YFP images were acquired simultaneously for most of the experiments. Sequential acquisition of CFP and YFP channels in alternative orders were tested and gave the same result as simultaneous acquisition.

**FRAP and photomanipulation**. For FRAP experiment, ROCK-GFP, Flw-YFP, and MyoII-GFP at the medial basal cortical regions were photobleached at around 4.2 micron size (488 nm laser at 100% power, 40–50 iterations) using a Zeiss LSM710 confocal microscope with ×40, numerical aperture 1.3 inverted oil lens. Following photobleaching, confocal images were acquired at the plane of medial basal cortex every 2 or 15 s.

For photoexcitation experiment, live-cell imaging was performed using a Zeiss LSM710 confocal microscope with ×40, numerical aperture 1.3 inverted oil lens, with a 488-nm argon laser and a 568-nm green HeNe laser. LARIAT clustering system was effectively induced by the blue light wavelength (400–510 nm), and thus here 488-nm argon laser was used to do the photoexcitation. To avoid the strong photobleaching effect on both GFP and RFP signals during photoexcitation, the 488 nm laser was set at 6% power level to do both GFP signal scanning and pulsed photoexcitation of LARIAT optogenetic system in a time-lapse imaging acquisition taken every 30 s.

**Drug treatments**. Egg chambers were dissected in live imaging medium, and then incubated with Rho inhibitor Rhosin[39] (Merk) at 250 µM for 20 min before being mounted for imaging.

**Image processing and data analysis**. Images were processed with MATLAB and Image J. For all images, the background (intensity of area without sample) was subtracted.

Image J were used to calculate the intensity of an individual cell as the average value of all pixels within the cell area. In time-lapse experiments, images were processed by MATLAB to correct photobleaching automatically. For dual-color imaging, the intensities were calculated from manually outlined cell areas if membrane-fluorescent protein was not present to mark cell boundaries.

For Rho FRET image, CFP and YFP images were first processed by ImageJ software. A background region of interest was subtracted from the original image. The YFP images were registered to CFP images using the TurboReg plugin. Gaussian smooth filter was then applied to both channels. The YFP image was thresholded and converted to a binary mask with background set to zero. The final ratio image was generated with the MATLAB program, during which only the unmasked pixel was calculated[38]. To determine the FRET signal ratio between basal junction and medio-basal region, each follicle cell was separated into junctional region and medial region (based on the Myo-II distribution region in control cells), and then the FRET signals in either region were analyzed using MATLAB.

The distribution of oscillation periods was generated by measuring the intervals between each pair of adjacent peaks. We applied autocorrelation to calculate the period of time series with different time offsets. This method averages out irregularities in the sequence and gives a similar average period. We found that autocorrelation was more robust and provided better results in analyzing irregular signals with a small amplitude, in some genetic backgrounds with strong reduction of Myo-II intensity compared with control[9]. To produce the average curves which show the dynamic changes of medial basal mean Myo-II intensity in Fig. 6d, h, l, p, t, x, we first synchronized all oscillating cycles, from different follicle cells with the same genetic backgrounds in several egg chambers, before this calculation. Each pink curve is the dynamic change from the average of Myo-II intensities in different cells with a range marked by pink color.

To analyze the dynamics of ROCK-GFP, Flw-YFP, and MyoII-GFP, we used the methods proposed from EMBL *FrapCalc* software for quantitative analysis of frap data (the software is available at https://github.com/framasoft/framacalc). The detailed analysis method has been outlined at Supplementary Information.

About the analysis of the ROCK-GFP and MyoII-RFP flow movements, there are two class of methods to estimate motion without analysis of single molecule fluorescent trajectories. The first class of methods is based on direct 2D cross correlation (in space domain or frequency domain) end are used on Particle Intensity Velocimetry (PIV) or in spatio-temporal correlation analysis like STICCS[40]. These methods are insensitive to noise from the image, because noise is an uncorrelated information frame by frame. But the methods need a good window variable size analysis and we had some difficulty to find a good estimation of sub pixel motion (due to interpolation method of 2D correlation peak in the presence of noise). We thus found that using the other family of classical motion estimation, based on optical flow analysis, provided a better read-out. We used the classical Horn–Schunck algorithm[40,41] to estimate the flow of myosin from open source image analysis platform (ICY software). The algorithm assumes smoothness in motion over the area of analysis and converges to solutions which show more smoothness. But it is more sensitive to noise than 2D cross correlation methods and the restauration of the image need to be done before. Thus, a 3D median filter with the kernel size close to PSF size has been performed before the estimation of the optical flow. This step reduced a lot the sensitivity of the Horn-Schunk methods to salt and pepper noise from confocal microscopy. Furthermore, an equalization of the histogram was performed to reduce fluctuation of intensity frame by frame. The largely known contrast limited adaptive histogram equalization (CLAHE) method has been used from the open source image analysis platform (ICY software)[42].

Tissue elongation was measured by the AP to DV length ratio of S9 and S10 egg chambers.

For quantification of cluster formation during photoexcitation of LARIAT system in *Drosophila* follicle cells, all time series of GFP images were automatically adjusted by MATLAB to the initial image intensity in order to reduce the effect of photobleaching. Then background noise was subtracted from the adjusted images before the measurement of clusters. Clusters were defined as discrete puncta of fluorescence with criteria of fluorescence intensity (1200–4095 arbitrary units), size (>0.2 µm$^2$), and circularity (0.35–1.0 arbitrary units). The area of clusters per cells was measured with MATLAB.

**Immunohistochemistry**. *Drosophila* ovaries were dissected in Schneider's medium and fixed with 4% formaldehyde for 20 min for most of antibodies, except of p-MLC, in which samples were fixed with 8% formaldehyde. After fixation, the egg chambers were rinsed with PBST (PBS with 0.3% Triton X-100) three times. The egg chambers were incubated with various first antibodies normally overnight in cold room. Antibody staining was performed as described previously. Anti-ROCK antibody (rabbit, 1:1000 dilution) was produced in this study. Anti-MBS antibody (rabbit, 1:400 dilution) was from Change Tan[24]. Anti-Armadillo antibody (mouse

**Fig. 7** Basal Myo-II oscillations are inhibited by the light-induced ROCK-GFP clustering while enhanced by Flw-YFP clustering. **a**, **b** Basal views of ROCK-GFP/p-MLC signal (**a**) and Flw-YFP/p-MLC signal (**b**) marked by p-MRLC antibody staining in the LARIAT-expressing follicle cells, under the dark condition or after 1 h photoexcitation by visible light. Both scale bars are 10 µm. **c**, **d** Quantifications of average GFP/YFP clustering area (**c**) and relative p-MRLC intensity (**d**) before and after photoexcitation in the LARIAT-expressing follicle cells. **e**, **f** Time-lapse series of the representative LARIAT-expressing follicle cells, labeled with ROCK-GFP and MyoII-mCherry (**e**) and Flw-YFP and MyoII-mCherry (**f**), under blue light excitation. Both scale bars are 5 µm. **g**, **h** Quantifications of the dynamic change of relative Myo-II intensity and relative area of ROCK clusters (**g**) and Flw clusters (**h**) in the LARIAT-expressing follicle cells, under photoexcitation. **i**, **j** Quantifications of relative Myo-II intensity (**i**) and average GFP/YFP clustering area (**j**) before and after photoexcitation in the LARIAT-expressing follicle cells. **k** Quantification of oscillating cycle time period of Myo-II signals in the LARIAT-expressing follicle cells, labeled with MyoII-mCherry together with either ROCK-GFP or Flw-YFP, before and after photoexcitation. *n* is the number of samples analyzed. Error bars indicate ±s.d. *p* < 0.001 means significant difference by Student's *t*-test

N27A1, 1:50 dilution) was from the Developmental Studies Hybridoma Bank. Anti-p-MLC (rabbit, 1:10 dilution) was from Cell Signaling (catalogue number 3671). Secondary antibodies conjugated with Alex-488 and Alexa-561 (Molecular Probes) were used in 1:400 dilutions. Samples were imaged on a Zeiss 710 or SP8 confocal microscope.

**Statistics.** All data are presented as mean ± standard deviation. Statistical analysis to compare results among groups was carried out by Student's $t$-test with two distribution tails. A value of $P < 0.05$ was considered to be statistically significant, while a value of $P < 0.001$ was considered to be highly statistically significant.

**Code availability.** The codes used for analyses of different images (including FRET and LARIAT) are available from the corresponding author on reasonable request.

**Data availability.** The data sets generated during and/or analysed during the current study are available from the corresponding author on reasonable request.

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

## Acknowledgements

We thank Adam Martin, Yohanns Bellaiche, Daniel Kiehart, Bloomington *Drosophila* stock center and Vienna *Drosophila* RNAi center for flies. We thank Change Tan for MBS antibody. We thank Tri Toulouse platform for imaging acquisition advice. We thank Francois Schweisguth and Karine Belguise for discussion of manuscript preparation. This work was supported by the Institut National de la Santéet de la Recherche Médicale [the ATIP-Avenir program (2012–2016)]; Région Midi-Pyrénées Excellence program (2013–2016); the National Natural Science Foundation of China (grant number 81670552 to B.Y., 11772088, 31470906 to Y.L.); the National Science Foundation of USA (grant number NSF 1456053 to J.A.M.); and the Indian Foundation (DST [SB/YS/LS-88/2014] and UGC [MRP-MAJOR-ZOOL-2013-16812] to P.M.

## Author contributions

X.Q. and C.L. performed image acquisition and transgene analysis. X.Q. and J.L. processed and analyzed images. E.H. did the mathematical modeling. T.M. conducted the data analysis of FRAP experiments. V.C.-C. made the constructs for transgenic flies. P.M. and J.A.M. made the antibody for ROCK. X.Q., E.H., T.M., Y. L., B.Y., and X.W. prepared the manuscript. X.Q., E.H. and X.W. designed the experiments. All authors participated in the interpretation of the data and the production of the final manuscript.

## Additional information

**Competing interests:** The authors declare no competing interests.

