## [Peer Review File(PDF 803 kb) · Nature Communications]

Reviewers' comments:

Reviewer #1 (Remarks to the Author):

In this manuscript " A biochemical network controlling basal myosin oscillation", the authors use a combination of genetic manipulations and drugs treatments coupled with advanced microscopy techniques to collect data about the oscillations of basal myosin II concentration in the *Drosophila* ovarian epithelium (during a certain stage of developpement).

This data is understood with a simple reaction-diffusion model which is also used to motivate some experiments. The aim of the paper is to support a bio-chemical explanation of the oscillations relying on the analysis of the intricate regulation pathway for Myosin II activation. The main idea advocated by the paper is that the drift induced by myosin contractility is not the driving force of the oscillations but that these oscillations rely on a chemical oscillator originating from the activation/inhibition relations between three players of the contractility regulation pathway: Rock, MBS/flw and Myosin II.

This idea is interesting, well explained and well supported by experiments. This paper is an important building block which will shed new light on the controversial explanations related to the funademental pheonomenon of myosin II oscillations. However, the theoretical model contains some problems/lack of precision which need to be addressed before i can recommend the publication of the paper.

I list below the main problems i have found reading this manuscript :

Major problems:

1) In paragraph 1.4, the condition for a positive flux $1+B-A$ contradicts the oscilations condition $A > 1+2B$. This may not be important for the 3 specie model but it raises big doubts for the reader. In particular, one starts to wonder if, assuming a negative flux (necessary for the oscillations), some concentrations could not become negative. My feeling is that this point should be made cristal clear for the paper not to lose impact. A phase diagram in the (A,B) space or (A, τ_m/τ_R) space would be needed to clarify the situation.

2) The spatial case of the traveling pulse is not clearly explained and analysed. It is not clear from where the pulse speed emerges. The type of the non-linearity is fundamental in the propagation of a wave in the FKPP case so one cannot really argue using this case if the non-linearity is different (as stated in the text). If there is an analogy, it sould really be traced or at least, a reference should be given. It is also unclear how the peak would dissociate at the center of the cell (where a zero flux condition should apply). 2D numerical simulations of system (20) on a disc would also be in order.

Minor problems:

1) It would be helpfull if the authors could spacificy what they mean by the words 'active' and 'inactive' and 'interaction' line 53/54. Myosin is both a cross-linker and a molecular motor, so the interaction with the F-actin could mean detachment/attachment of the motor or generation of power stroke in the meshwork.

In the same vein, line 146, what 'dominant negative' means ? Is myosin still attached and only power stroke is inhibited or myosin is detached ?

In the concluision, it would be helpful to give a final view about this: Should we understand that all the oscillations of myosin II concentration is happening on detached myosin and that some transient attachement to the cytoskeleton has very little impact on the total myosin dynamic ? Or the picture is different and, if myosin is attached, why the self amplifying drift

term stemming from myosin contractility is negligible in the myosin II dynamic (compared to exchange with the medial pool for instance) ?

Maybe all this clear for specialists but it was not for some general audience like me.

2) line 229. Aren't there only 4 non-dimensional parameters (when time is non dimensionalized) ?

3) line 237 the recovery times do not seem to be "very different"

Theory supplementary note:

4) I would drop the generality in 'the most general chemical equations'. There could be time delays coming for instance from an additional specie playing the role of a hidden variable.

5) several proposed model(s)

6) look->loop

7) The auto-activation of R in the model is represented by a linear term. Generally, it is rather accounted using a quadratic term. Any special reason ?

8) The inhibition of MBS by rock could be accounted for in the equation for dM/dt and not on the equation for dm/dt as suggested in the text (btw a τ_m is missing to divide J in this equation beginning of page 3, no ?). Maybe this would play a more important role in that case.

9) avoid using 'steady state'. I think what you mean would be more clear using the term 'homogeneous solution'.

10) In paragraph 1.5, i think it would be much more direct and clear to exhibit the spectrum of M instead of this identification of characteristic polynoms. The instability case where all three eigenvalues are real but one (at least) is positive could also be treated this way. Operating with this ansatz leaves the reader a bit in doubt as to the oscillations are the generic instability arising or if something different could happen in the phase space. Again, a phase diagram would clarify this.

11) $\partial_n M$ (after equation 20). Also the meaning of the differential wrt to the normal of the domain of computation should be given.

Paper construction:

1) I think the model plays a pivotal role in this paper and it would be nice to have equation (20) in the main text. For a physicist, this equation is the key of the mechanism.

2) For non-specialists, it is usefull to know, with all these fluorescent marking, what is the proportion of times when cells die before an experiment actually 'works'. But this is just a general side note.

Reviewer #2 (Remarks to the Author):

Qin and colleagues investigate the basal actomyosin oscillations that shape oocytes in the follicular epithelial cells of the Drosophila egg chamber. They find that ROCK and MBS, the kinase and phosphatase, respectively, that regulate myosin activation, switch their localization from apical to

basal during the stages of oogenesis in which basal oscillations are most prominent. Genetic activation of ROCK and inhibition of MBS result in long eggs along the AP axis, while deactivation of ROCK and hyperactivation of MBS result in rounder eggs. The authors find that Rho localizes to cell interfaces on the basal surface of follicular epithelial cells, with little oscillatory activity. In contrast, ROCK oscillates on the medial-basal surface, with ROCK peaks preceding myosin by ~ 1 min, and both MBS and Flw (the two subunits of the myosin phosphatase) following myosin by ~ 1 min. Inhibiting actomyosin contractility by overexpressing a dominant-negative myosin heavy chain suppressed the changes in cell area associated with basal myosin oscillations, but surprisingly it did not disrupt the myosin oscillations themselves. Using image velocimetry (optical flow) the authors find that ROCK flows from junctions, where Rho resides, to medial regions, in a Rho-dependent process. The authors find that ROCK levels on the basal cell surface are negatively regulated by MBS and positively regulated by ROCK. Based on the known relationships between ROCK, MyoII, and MBS/Flw, the authors create a mathematical model that is able to qualitatively predict the relative dynamics of these proteins on the basal cell surface. The model predicts that ROCK activity correlates directly with the period of oscillation, and MBS activity is indirectly correlated. The authors verify these predictions experimentally. Using FRAP analysis, the authors show that ROCK and Flw flow from the membrane towards the interior of the cells, while MyoII remains mainly cytoplasmic. When the diffusion rates of these molecules (calculated by FRAP) are included in the model, the authors obtain a wave propagating from the cell edge to the center at a constant velocity, as shown by their data. Finally, the authors apply an optogenetic method that they develop to induce clustering and inactivation of GFP-tagged proteins, and show that ROCK inactivation suppresses oscillations (shorter amplitude and cycle time), and Flw or MBS inactivation enhances oscillations (increased amplitude and period), thus confirming both genetic and modelling analyses.

I enjoyed reading the manuscript and I think it will be of general interest to the morphogenesis community, specially if the authors can address a couple of experimental questions that I have. Oscillatory actomyosin networks have now been observed in an number of systems and animals, but the logic of their dynamics is still unclear. The authors use multiple methods and provide new information on the mechanisms of cytoskeletal oscillation. I find it unfortunate that the methods, are buried in the supplement. Also, I have some questions about the timing and reagents used for some experiments. Therefore, I propose to address the following points:

MAJOR

1. In Figure 1e-j, the authors show the effects of changing ROCK and MBS activity on egg shape using changes in shape from stage 10 to stage 14 of *Drosophila* oogenesis. However, they mention that basal oscillations are active between stages 9 and 10B. Thus, they author should show effects on egg shape at those two stages, rather than comparing stage 10 and 14.

2. The authors use Myo-II EE mutants to show that myosin filament formation does not promote ROCK recruitment to the basal surface, but rather inhibits it. The assumption here is that Myo-II EE mutants have an increased ability to form filaments with respect to wild-type Myo-II. However, recent evidence (Vasquez, Heissler, Billington, et al., eLife, 2016) shows that Myo-II EE in *Drosophila* does not mimic the phosphorylated state, and in fact, even though Myo-II EE molecules can form filaments, those filaments are disassembled in the presence of ATP. In addition, Myo-II EE displays reduced motor activity. Therefore, the Myo-II EE scenario could represent a myosin mutant with reduced ability to form filaments to scaffold ROCK, or to generate contractile forces that may direct ROCK flows from the membrane towards the interior of the cells. What is the effect on ROCK levels of Myo-II AA, which should inhibit filament formation? Is it similar to Myo-II EE or the opposite?

3. At the end, I am left with a big question: why does ROCK shuttle medially? The authors should at least discuss some explanation for this, specially if they claim (see Discussion) that there is no advection. Isn't it possible that the initial contraction of a few medial motors (since myosin is mostly medial and not membrane associated) triggers the flow of ROCK, and positive feedback

between ROCK and myosin reinforces the medial-basal recruitment of myosin? Understanding the effects of Myo-II AA mutants on ROCK medial levels would help determine if additional interactions are necessary in Figure 5b and the model.

MINOR

1. The authors use optical flow to quantify ROCK movement between junctional and medial regions. I have two questions about this:

1a. Line 170: the authors reference the Methods for the optical flow analysis, but I could not find any reference to optical flow in the Methods.

1b. I was confused by Figure 4c. I understand that the grayscale images are heavily-smoothened versions of the corresponding channel in the color images? If so, what were the type and parameters of smoothing applied?

2. In the abstract, the authors should refer to "myosin II", not "myosinII".

3. What do the n numbers indicate in Figure 1c-d? Is that number of cells, egg chambers, something else?

4. In Figures 2j, k and their captions, what do the authors mean by local Flw-YFP? Are local and partial the same? If so, use only one term to refer to both.

5. In Figure 6a'-c' the authors may want to use the same color scheme used in the other panels, with controls in blue and cells expressing the transgenes in red.

6. In Supplementary Fig. 4b, why do the authors change the color scheme for the different regions from ROCK to MyoII? Doesn't it make more sense to color-code the regions consistently (e.g. red is always the external circle, green is always the internal one)?

7. Line 145: "of cytoskeleton" should be "of the cytoskeleton".

8. Line 692: rather than "interacts with" the authors may want to use "is in close proximity to", as they show no evidence for direct interactions between Rho and ROCK.

9. Line 702: "the number percentage" should be "the percentage".

10. Line 703: again, "the number percentage" should be "the percentage".

Reviewers' comments:

Reviewer #1 (Remarks to the Author):

In this manuscript " A biochemical network controlling basal myosin oscillation", the authors use a combination of genetic manipulations and drugs treatments coupled with advanced microscopy techniques to collect data about the oscillations of basal myosin II concentration in the Drosophila ovarian epithelium (during a certain stage of developpement). This data is understood with a simple reaction-diffusion model which is also used to motivate some experiments. The aim of the paper is to support a bio-chemical explanation of the oscillations relying on the analysis of the intricate regulation pathway for Myosin II activation. The main idea advocated by the paper is that the drift induced by myosin contractility is not the driving force of the oscillations but that these oscillations rely on a chemical oscillator originating from the activation/inhibition relations between three players of the contractility regulation pathway: Rock, MBS/flw and Myosin II. This idea is interesting, well explained and well supported by experiments. This paper is an important building block which will shed new light on the controversial explanations related to the funademental pheonomenon of myosin II oscillations. However, the theoretical model contains some problems/lack of precision which need to be addressed before i can recommend the publication of the paper.

I list below the main problems i have found reading this manuscript :

We thank the reviewer for his positive assessment of our manuscript, as well as for his very careful reading and constructive remarks, which we address point by point below.

Major problems:

1) In paragraph 1.4, the condition for a positive flux $1+B-A$ contradicts the oscillations condition $A > 1+2B$. This may not be important for the 3 specie model but it raises big doubts for the reader. In particular, one starts to wonder if, assuming a negative flux (necessary for the oscillations), some concentrations could not become negative. My feeling is that this point should be made cristal clear for the paper not to lose impact. A phase diagram in the (A,B) space or (A, τ_m/τ_R) space would be needed to clarify the situation.

Answer: We thank the reviewer for his careful reading of our manuscript and of its associated Supplementary Note. He is correct indeed that the positive flux condition (which we intended to state for the 3 species), does not allow for oscillations in the 2 species model. We thus clarified this point in the revised Supplementary Note (page 4), which would further argue for the need to consider a third variable in this problem.

Furthermore, as we make clearer in this revised version of the manuscript, we had systematically checked to the fact that concentrations do not become negative in our 3-species model (neither the non-spatial and spatial models - both 1D and 2D versions, as detailed below), a point that we now make clearer by adding numerical integrations for different regions of the (A,B) phase diagram, see Supple Fig. 5, and by referring to these specifically in the main text. As detailed below, we also added (A,B) phase diagram for the case of ROCK feedbacking on MBS, as suggested by the referee. Finally, the referee is right however than in the two-species model of 1.4.1, one would need additional non-linearity to generically avoid negative concentrations in all parts of the phase diagram, a point we discuss in more depth (page 4).

2) The spatial case of the traveling pulse is not clearly explained and analysed. It is not clear from where the pulse speed emerges. The type of the non-linearity is fundamental in the propagation of a wave in the FKPP case so one cannot really argue using this case if the non-linearity is different (as stated in the text). If there is an analogy, it should really be traced or at least, a reference should be given. It is also unclear how the peak would dissociate at the center of the cell (where a zero flux condition should apply). 2D numerical simulations of system (20) on a disc would also be in order.

Answer: Firstly, we fully agree with the referee that a 2D simulation on a disk is important to prove that the results in 1D still hold, in particular on peak dissociation. We implemented this, using a Finite Element Method approach (via FreeFem), and adding the same boundary conditions as in 1D (at $x=0$) around the disk (or alternatively, a Rock activation preferentially around only one angle $\theta=0$ of the disk edge). Importantly, this fully confirmed our 1D simulations (which were done using Mathematica), both on the period of the oscillation (Supple Fig. 5i-k), and with preferential Rock activation at the entire rim resulting in an inwards kinematic wave – or preferential Rock activation at one side of the rim resulting in a biased kinematic wave (see Supple Fig. 5i,j for snapshots and curves of a time evolution of the simulation in each case). In the first case, to answer the specific question of the reviewer, we do observe in particular peak dissociation in the center of the disk, which is due to MBS/flw “catching up” to MyoII, in the same way as what happens in the non-dimensional version of the model.

Secondly, we agree with the reviewer that the analogy with Fisher-KPP might be confusing here (initially developed from the quadratic nature of the non-linearity, and as the state $R=0$, m/M finite is unstable while the state $R=M=m=1$ is), due to the fact that one impose “boundary” biases in this problem, so that the wave is simply a sole consequence of Fisher-KPP. We thus removed the reference, and

discussed further instead the 2D simulation and peak dissociation in this context (p10 of the updated Supplementary Note).

Minor problems:

1) It would be helpful if the authors could specify what they mean by the words 'active' and 'inactive' and 'interaction' line 53/54. Myosin is both a cross-linker and a molecular motor, so the interaction with the F-actin could mean detachment/attachment of the motor or generation of power stroke in the meshwork.

Answer: Both have been reported to be dependent on the phosphorylation of MyoII (Dawes-Hoang et al, 2005), a statement we made more precise, and we added this reference in the main text.

In the same vein, line 146, what 'dominant negative' means? Is myosin still attached and only power stroke is inhibited or myosin is detached?

Answer: Dominant negative mutations result in a partially altered molecular function. In the case of dominant-negative zipper (name of MHC in *Drosophila*), this will affect the MyoII heavy chain, which decreases both the power stroke and the binding to F-Actin. The fact that the power stroke is strongly inhibited is confirmed by the lack of area changes in these cells, although the presence of oscillations point indeed to some conserved binding ability, a point we now discuss in the Discussion section.

In the conclusion, it would be helpful to give a final view about this: Should we understand that all the oscillations of myosin II concentration is happening on detached myosin and that some transient attachment to the cytoskeleton has very little impact on the total myosin dynamic? Or the picture is different and, if myosin is attached, why the self amplifying drift term stemming from myosin contractility is negligible in the myosin II dynamic (compared to exchange with the medial pool for instance)? Maybe all this clear for specialists but it was not for some general audience like me.

Answer: We amended the last paragraphs to provide more discussion about the DN-zipper experiment, which we believe helps to clarify this, as it shows that the force generation related to the power-stroke of MyoII on F-Actin is not necessary for the dynamics. This being said, it remains hard to test whether direct linkage is necessary, or what is the exact dynamics of binding/unbinding to the cytoskeleton, although we note that recovery of MyoII intensity after FRAP comes from exchange with the cytoplasmic pool rather than by 2D diffusion

along the cortex, arguing that exchange with the medial pool plays a key role at the time scale of the oscillation.

2) line 229. Aren't there only 4 non-dimensional parameters (when time is non-dimensionalized) ?

Answer: Indeed, the referee is correct that if time is non-dimensionalized, there would be only 4 non-dimensional parameters. However, in the rest of the article, we wished to make prediction on the absolute values of the period and delays (and not just their ratio), so that one cannot non-dimensionalize time.

3) line 237 the recovery times do not seem to be "very different"

Answer: We agree with the referee and deleted this subjective mention of « very » in the text.

Theory supplementary note:

4) I would drop the generality in 'the most general chemical equations'. There could be time delays coming for instance from an additional specie playing the role of a hidden variable.

Answer: We thank the referee for the good suggestion, and made our statement more precise by specifying that we are looking at ODEs without time delays (p1 of the updated Supplementary Note).

5) several proposed model(s)

6) look->loop

Answer: We thank the referee for noticing these typos, and corrected them.

7) The auto-activation of R in the model is represented by a linear term. Generally, it is rather accounted using a quadratic term. Any special reason ?

Answer: As we now made more precise in the revised version of the Supplementary Note (bottom of the first page), we followed here a convention take from the classical FitzHugh-Nagumo model, in which auto-activation is represented as a linear term. We used FitzHugh-Nagumo as a benchmark for the model, as it is both a paradigm example for temporal oscillations (in 0D models), and spatio-temporal patterns in the presence of diffusion (in spatial models), making it particularly topical for the phenomena we examine.

8) The inhibition of MBS by rock could be accounted for in the equation for dM/dt and not on the equation for dm/dt as suggested in the text (btw a τ_m is

missing to divide J in this equation beginning of page 3, no ?). Maybe this would play a more important role in that case.

Answer: We agree on the importance to discussing this further. We thus carried out the analysis on including a term " $-\alpha R$ " in the dM/dt equation, as the referee suggests (page 3). This does not qualitatively change the transition from homogeneous to oscillatory solutions described in the main text (for $A > A_{c1}(B)$), although it creates a secondary zone of non-oscillatory instability for $A < A_{c2}(B)$, characterized by a single real eigenvalue becoming positive, see Supple Fig. 5I for a corresponding (A,B) phase diagram. This confirms that such a feedback would not be relevant for the types of periodic solutions we are looking at. We discuss this (p3) in further depth in the revised Supplementary Note.

9) avoid using 'steady state'. I think what you mean would be more clear using the term 'homogeneous solution'.

Answer: We thank the reviewer for the suggestion and have modified the text throughout the Supplementary Note accordingly.

10) In paragraph 1.5, i think it would be much more direct and clear to exhibit the spectrum of M instead of this identification of characteristic polynoms. The instability case where all three eigenvalues are real but one (at least) is positive could also be treated this way. Operating with this ansatz leaves the reader a bit in doubt as to the oscillations are the generic instability arising or if something different could happen in the phase space. Again, a phase diagram would clarify this.

Answer: We thank the referee for the good suggestion: although the spectrum of M cannot be written analytically, we went back to the phase diagram and checked systematically for other behavior for the eigenvalues (such as a single real part being positive without imaginary parts as the referee suggests), but found that the oscillatory case was the only one present in the phase diagram. We state this explicitly now in the Supplementary Note (in paragraph 1.6). As detailed above, we also plotted three examples of numerical solutions in three different regions of the phase diagram, for illustration of this (Supple Fig. 5e). We also did the same (see response above) in the case of a direct feedback of ROCK on MBS/flw, which does exhibit a more complex phase diagram (Supple Fig. 5I)

11) $\partial_n M$ (after equation 20). Also the meaning of the differential wrt to the normal of the domain of computation should be given.

Answer: We thank the referee for pointing this out.

Paper construction:

1) I think the model plays a pivotal role in this paper and it would be nice to have equation (20) in the main text. For a physicist, this equation is the key of the mechanism.

Answer: We followed the advice of the reviewer and added Equation 20 to the main text.

2) For non-specialists, it is useful to know, with all these fluorescent marking, what is the proportion of times when cells die before an experiment actually 'works'. But this is just a general side note.

Answer: All our experiments have been carried out under the live cell imaging condition (except for antibody staining, such as ROCK ab and MBS ab results). For all fluorescence dynamics, FRAP and optogenetics, the laser powers we used for live cell imaging are under healthy conditions for epithelial cells and tissue, within at least 1-2 hours of live tissue culture ex vivo. Thus, no death of cells has been observed in our live cell experiments.

Reviewer #2 (Remarks to the Author):

Qin and colleagues investigate the basal actomyosin oscillations that shape oocytes in the follicular epithelial cells of the *Drosophila* egg chamber. They find that ROCK and MBS, the kinase and phosphatase, respectively, that regulate myosin activation, switch their localization from apical to basal during the stages of oogenesis in which basal oscillations are most prominent. Genetic activation of ROCK and inhibition of MBS result in long eggs along the AP axis, while deactivation of ROCK and hyperactivation of MBS result in rounder eggs. The authors find that Rho localizes to cell interfaces on the basal surface of follicular epithelial cells, with little oscillatory activity. In contrast, ROCK oscillates on the medial-basal surface, with ROCK peaks preceding myosin by ~ 1 min, and both MBS and Flw (the two subunits of the myosin phosphatase) following myosin by ~ 1 min. Inhibiting actomyosin contractility by overexpressing a dominant-negative myosin heavy chain suppressed the changes in cell area associated with basal myosin oscillations, but surprisingly it did not disrupt the myosin oscillations themselves. Using image velocimetry (optical flow) the authors find that ROCK flows from junctions, where Rho resides, to medial regions, in a Rho-dependent process. The authors find that ROCK levels on the basal cell surface are negatively regulated by MBS and positively regulated by ROCK. Based on the known relationships between ROCK, MyoII, and MBS/Flw, the authors create a mathematical model that is able to qualitatively predict the relative dynamics of these proteins on the basal cell surface. The model predicts that ROCK activity correlates directly with the period of oscillation, and MBS activity is indirectly correlated. The authors verify these predictions experimentally. Using FRAP analysis, the authors show that ROCK and Flw flow from the membrane towards the interior of the cells, while MyoII remains mainly cytoplasmic. When the diffusion rates of these molecules (calculated by FRAP) are included in the model, the authors obtain a wave propagating from the cell edge to the center at a constant velocity, as shown by their data. Finally, the authors apply an optogenetic method that they develop to induce clustering and inactivation of GFP-tagged proteins, and show that ROCK inactivation suppresses oscillations (shorter amplitude and cycle time), and Flw or MBS inactivation enhances oscillations (increased amplitude and period), thus confirming both genetic and modelling analyses.

I enjoyed reading the manuscript and I think it will be of general interest to the morphogenesis community, specially if the authors can address a couple of experimental questions that I have. Oscillatory actomyosin networks have now been observed in an number of systems and animals, but the logic of their dynamics is still unclear. The authors use multiple methods and provide new information on the mechanisms of cytoskeletal oscillation. I find it unfortunate that the methods, are buried in the supplement. Also, I have some questions about the timing and reagents used for some experiments. Therefore, I propose to address the following points:

We thank the reviewer for his positive assessment of our manuscript, as well as for his very careful reading and constructive remarks, which we address point by point below.

MAJOR

1. In Figure 1e-j, the authors show the effects of changing ROCK and MBS activity on egg shape using changes in shape from stage 10 to stage 14 of *Drosophila* oogenesis. However, they mention that basal oscillations are active between stages 9 and 10B. Thus, they author should show effects on egg shape at those two stages, rather than comparing stage 10 and 14.

Answer: We thank the reviewer for this thoughtful suggestion, which we implemented in the revised version of our manuscript. During oogenesis, stage 9 is composed of different stages, including early S9, middle stage 9 and late stage 9. Basal myosin oscillation becomes gradually stronger from early stage 9 to late stage 9. To characterize the effect of changing ROCK and MBS activity on egg shape with high temporal accuracy, we thus collected images and quantified the tissue shape at these 3 different stage 9 periods, in addition to stage 10. When myosin activity is inhibited (either by ROCK RNAi or MBS N300 expression), tissue elongation cannot occur, and tissue shape at S9 is similar to that at S10. Conversely, when myosin activity is enhanced (either by ROCK CA or MBS RNAi), tissue elongation gradually increases from early stage 9 to stage 10. This results in significantly more elongated oogenesis, validating our previous analysis. All these results have been updated in our Fig. 1e, f. Finally, we deleted the figure of S14 in our figure 1 to follow the suggestion of the reviewer.

2. The authors use Myo-II EE mutants to show that myosin filament formation does not promote ROCK recruitment to the basal surface, but rather inhibits it. The assumption here is that Myo-II EE mutants have an increased ability to form filaments with respect to wild-type Myo-II. However, recent evidence (Vasquez, Heissler, Billington, et al., eLife, 2016) shows that Myo-II EE in *Drosophila* does not mimick the phosphorylated state, and in fact, even though Myo-II EE molecules can form filaments, those filaments are disassembled in the presence of ATP. In addition, Myo-II EE displays reduced motor activity. Therefore, the Myo-II EE scenario could represent a myosin mutant with reduced ability to form filaments to scaffold ROCK, or to generate contractile forces that may direct ROCK flows from the membrane towards the interior of the cells. What is the effect on ROCK levels of Myo-II AA, which should inhibit filament formation? Is it similar to Myo-II EE or the opposite?

Answer: In this revised manuscript, we now specifically examined the case of Myo-II AA, to address the question of the reviewer.

To address the potential effect of filament formation on ROCK, we first checked the effect of Myo-II WT, Myo-II EE and Myo-II AA on Myo-II RFP accumulation (in clones). From the following figure, we can clearly see that Myo-II AA strongly reduces the stress fibers formation, while Myo-II EE enhances the stress fibers, compared with Myo-II WT clonal cells. Consistently, when we checked the effect of Myo-II WT, Myo-II AA, and Myo-II EE overexpression on S10 tissue shape, we found that Myo-II EE-expressing tissue got significantly more elongated than normal tissue, while Myo-II AA-expressing tissue got less elongated, and Myo-II WT-expressing tissue is similar to normal control tissue.

(We didn't include this figure in our revised manuscript, since we felt our data of ROCK-signal by antibody/ROCK-GFP is already clear enough for our conclusion; if reviewer think it is important for this figure to furthermore support our hypothesis, we can add it as one supplementary figure)

We then checked the effect of Myo-II AA on endogenous ROCK basal medial accumulation by ROCK antibody staining, we found that Myo-II AA clonal cells have the same endogenous ROCK basal medial accumulation, as neighboring WT cells (added to revised manuscript as **Supple Fig.3f, g**). Importantly, we also obtained a similar result when we checked the dynamic ROCK-GFP in Myo-II AA clonal cells (please see the updated Fig.5a and Supple Fig. 3a, and the following figure—dynamic ROCK-GFP patterns). This confirmed our hypothesis that “myosin filament formation does not promote ROCK recruitment” (as Myo-II

AA, which strongly inhibits filament formation does not cause decreased ROCK levels).

We updated both ROCK antibody staining and ROCK-GFP data in our revised manuscript (updated Fig. 5a and Supple Fig. 3).

(This figure of ROCK-GFP dynamics has not been included in our revised manuscript, since in Fig.5a we have quantified the average intensity of medial basal ROCK signals to summarize the effect; if reviewers think it is important to include this dynamic figure, we can add it into one supplementary figure)

To further confirm whether actomyosin contraction force might direct ROCK flows from the membrane towards the interior of the cells, we used DN-zipper (dominant negative form of Myosin heavy chain, with a weak GFP-tag) clonal

cells to check the effect on endogenous ROCK basal medial accumulation by ROCK antibody staining. Consistent with Myo-II AA clonal cells, DN-zipper clonal cells have the same endogenous ROCK basal medial accumulation, as neighboring WT cells. We also updated this ROCK antibody data in current Supple Fig. 3. As shown in our Fig. 3, DN-zipper expression strongly blocks basal area reduction of follicle cells, indicating the loss of actomyosin contractions due to the failure of Myosin loading on F-actin. Thus, strong inhibition of actomyosin contractions by DN-zipper has no detectable effect on ROCK basal medial accumulation.

Taken together, we can conclude that contractile forces are not a critical factor to induce ROCK flows and thus ROCK basal medial accumulation.

3. At the end, I am left with a big question: why does ROCK shuttle medially? The authors should at least discuss some explanation for this, specially if they claim (see Discussion) that there is no advection. Isn't it possible that the initial contraction of a few medial motors (since myosin is mostly medial and not membrane associated) triggers the flow of ROCK, and positive feedback between ROCK and myosin reinforces the medial-basal recruitment of myosin? Understanding the effects of Myo-II AA mutants on ROCK medial levels would help determine if additional interactions are necessary in Figure 5b and the model.

Answer: We thank the reviewer for this question. As mentioned in the response above, we now added to this revised version that MyoII AA mutants display normal ROCK medial levels. This is consistent with the results we got from DN-zipper, which argue that motor contraction is dispensable for the oscillation and flow to occur. We added these results (Figure 5a, and supple Fig. 3) and discussed them further in the Discussion section of our revised manuscript. As to the question of how ROCK shuttle medially, we believe that this can be explained simply from its rapid 2D diffusion (assessed by FRAP in our manuscript). The capacity of ROCK to diffuse laterally along the lateral surface is in stark contrast with the absence of lateral diffusion of MyoII, which lead us to hypothesize that it is this diffusion of ROCK that couples its preferential Rho-mediated activation at the boundary to the activation of medial MyoII. In the revised manuscript, and as this was raised by Ref. 1 as well, we provide a 2D numerical integration of the model equations (Supple Fig.5i,j), which better demonstrate this diffusive flow.

MINOR

1. The authors use optical flow to quantify ROCK movement between junctional and medial regions. I have two questions about this:

1a. Line 170: the authors reference the Methods for the optical flow analysis, but I could not find any reference to optical flow in the Methods.

1b. I was confused by Figure 4c. I understand that the grayscale images are heavily-smoothened versions of the corresponding channel in the color images? If so, what were the type and parameters of smoothing applied?

Answer: There are two classes of methods to estimate motion without analysis single molecule fluorescent trajectories. The first class of methods is based on direct 2D cross correlation (in space domain or frequency domain) and are used on Particle Intensity Velocimetry (PIV) or in spatiotemporal correlation analysis like STICCS [1]. These methods are insensitive to the noise from the image because noise is an uncorrelated information frame by frame. But the methods need a good window variable size analysis and we had some difficulty to find a good estimation of sub pixel motion (due to interpolation method of 2D correlation peak in the presence of noise). We thus found that using the other family of classical motion estimation, based on optical flow analysis, provided a better read-out. We used the classical Horn–Schunck algorithm [1,2] to estimate the flow of myosin from open source image analysis platform (ICY software). The algorithm assumes smoothness in motion over the area of analysis and converge to solutions which show more smoothness. But it is more sensitive to noise than 2D cross correlation methods and restoration of the image need to be done before. Thus, a 3D median filter with the kernel size closed to the PSF size has been performed before the estimation of the optical flow. This step reduced a lot the sensitivity of the Horn-Schunck methods to salt and pepper noise from confocal microscopy. Furthermore, an equalization of the histogram was performed to reduce fluctuation of intensity frame by frame. The largely known Contrast Limited Adaptive Histogram Equalization (CLAHE) method have been used from of the open source image analysis platform (ICY software) [3].

[1] Hebert, B., Costantino, S., & Wiseman, P. W. (2005). Spatiotemporal image correlation spectroscopy (STICS) theory, verification, and application to protein velocity mapping in living CHO cells. *Biophysical journal*, 88(5), 3601-3614.

[2] Lecomte, T., Thibeaux, R., Guillen, N., Dufour, A., & Olivo-Marin, J. C. (2012, September). Fluid optical flow for forces and pressure field estimation in cellular biology. In *Image Processing (ICIP), 2012 19th IEEE International Conference on* (pp. 69-72). IEEE.

[3] Pujadas, A. B., Manich, M., Guillen, N., Olivo-Marín, J. C., & Dufour, A. C. (2016, April). Biophysical measurements in 2D and 3D live cell imaging using fluid dynamics and optical flow. In *Biomedical Imaging (ISBI), 2016 IEEE 13th International Symposium on* (pp. 980-983). IEEE.

All information of this method has been updated in our current version method sections and references have also been updated in our manuscript.

2. In the abstract, the authors should refer to "myosin II", not "myosinII".

Answer: We corrected this as suggested.

3. What do the n numbers indicate in Figure 1c-d? Is that number of cells, egg chambers, something else?

Answer: Thank reviewer for this unclear information. N in figure 1c-d is the number of cells from 4 independent egg chambers for each condition. We included this information in the updated figure legends.

4. In Figures 2j, k and their captions, what do the authors mean by local Flw-YFP? Are local and partial the same? If so, use only one term to refer to both.

Answer: Thank reviewer for this confusing information in our previous manuscript. We replaced the word "partial" in figure legends with "local", so it will be consistent with "local" used in figure captions.

5. In Figure 6a'-c' the authors may want to use the same color scheme used in the other panels, with controls in blue and cells expressing the transgenes in red.

Answer: We corrected this as suggested.

6. In Supplementary Fig. 4b, why do the authors change the color scheme for the different regions from ROCK to MyoII? Doesn't it make more sense to color-code the regions consistently (e.g. red is always the external circle, green is always the internal one)?

Answer: We corrected this as suggested. In addition, we noticed a couple of mistakes in the time frame of FRAP curve for both ROCK and myosin signals. And we also switched another sample for the FRAP analysis of ROCK, since our previous one has higher noise effect (the current one has much less noise for better view of FRAP result). All these corrected files have been updated in the current Supple Fig. 4.

7. Line 145: "of cytoskeleton" should be "of the cytoskeleton".

Answer: We corrected this as suggested.

8. Line 692: rather than "interacts with" the authors may want to use "is in close proximity to", as they show no evidence for direct interactions between Rho and ROCK.

Answer: We corrected this as suggested.

9. Line 702: "the number percentage" should be "the percentage".

Answer: We corrected this as suggested.

10. Line 703: again, "the number percentage" should be "the percentage".

Answer: We corrected this as suggested.

REVIEWERS' COMMENTS:

Reviewer #1 (Remarks to the Author):

The authors have fully addressed the concerns raised in my previous review. Overall, my impression is the paper quality and readability have improved. I can therefore recommend publication.

Below I list some optional issues that could be useful to the authors.

A nice addition to the paper could be made by predicting theoretically the velocity of the kinetic wave (Line 290) and its dependence on the kinetic coefficients of the model in the theory note. This may lead to further confirmation of the model validity.

-Line 66: I would remove 'mechanistically' as this paper precisely points at the fact the regulation is not necessary mechanical (i.e. dependant on macroscopic force)

-Line 187: 'in the absence of upstream regulators' could be changed to 'even in the absence of possible upstream regulators other than Rho1 that would be oscillatory'

-Line 217: 'These data hint' could be replaced by 'These data thus hint'

-Line 228: add a period at the end of the equations.

Say what are the parameters entering in the equation in physical terms. Maybe this can be coupled with the discussion after the equations by presenting the equations directly in dimensionless form.

-Line 245: You could give the corresponding estimates of the parameters entering your non-dimensional equation.

-Line 259: You could add the reference

-Line 275: You could say 'saturation effects in the kinetic coefficients entering in our simplified model'

Reviewer #2 (Remarks to the Author):

This was a well-done revision. I appreciate the effort done by the authors to address my concerns about the use of myosin-EE. I would include the two figures that the authors had in the response letter as supplemental figures in the manuscript, and I would remove all references to Myosin-EE (some data are still shown in Fig. 5a, but they are not discussed in the text). If the authors insist on keeping the Myosin-EE data in Fig. 5a, then it should be discussed in the text, in the context of the work by the Martin and Sellers lab published in eLife (which is currently not referenced).

REVIEWERS' COMMENTS:

Reviewer #1 (Remarks to the Author):

The authors have fully addressed the concerns raised in my previous review. Overall, my impression is the paper quality and readability have improved. I can therefore recommend publication.

We thank the reviewer for his positive assessment of our manuscript, as well as for his very careful reading and constructive remarks, which we address point by point below.

Below i list some optional issues that could be useful to the authors.

A nice addition to the paper could be made by predicting theoretically the velocity of the kinetic wave (Line 290) and its dependance on the kinetic coefficients of the model in the theory note. This may lead to further confirmation of the model validity.

Answer: On this point, although we agree with the reviewer that this would be interesting, the time step from the imaging does not allow for a very precise estimation of the velocity, especially given the fact that the basal surface is complex and spatially heterogeneous... We thus feel that this comparison might be hard to interpret, but discussed this question in more depth in the last sections of the Supplementary Theory Note.

-Line 66: I would remove 'mechanistically' as this paper precisely points at the fact the regulation is not necessary mechanical (i.e. dependant on macroscopic force)

Answer: We corrected this as suggested.

-Line 187: 'in the absence of upstream regulators' could be changed to 'even in the absence of possible upstream regulators other than Rho1 that would be oscillatory'

Answer: We corrected this as suggested.

-Line 217: 'These data hint' could be replaced by 'These data thus hint'

Answer: We corrected this as suggested.

-Line 228: add a period at the end of the equations.

Say what are the paramters entering in the equation in physical terms. Maybe this can be coupled with the discussion after the equations by presenting the equations directly in dimensionless form.

-Line 245: You could give the corresponding estimates of the parameters entering your non-dimensional equation.

Answer: We followed the suggestion and discussed each term right after the equation, and introduce the estimates for the characteristic times afterwards.

-Line 259: You could add the reference

Answer: We added the reference as suggested.

-Line 275: You could say 'saturation effects in the kinetic coefficients entering in our simplified model'

Answer: We corrected this as suggested.

Reviewer #2 (Remarks to the Author):

This was a well-done revision. I appreciate the effort done by the authors to address my concerns about the use of myosin-EE. I would include the two figures that the authors had in the response letter as supplemental figures in the manuscript, and I would remove all references to Myosin-EE (some data are still shown in Fig. 5a, but they are not discussed in the text). If the authors insist on keeping the Myosin-EE data in Fig. 5a, then it should be discussed in the text, in the context of the work by the Martin and Sellers lab published in eLife (which is currently not referenced).

Answer: Firstly, we included the two figures in our updated supplementary figure 3, and thus took some figures into new updated supplementary figure 4. So our original supplementary figures 4-8 have been updated as new supplementary figures 5-9. Secondly, we think that myosin-EE data is very important, and thus we feel that it is better to keep all results of myosin-EE in our final main text. We agreed with reviewer that myosin-EE data should be discussed in the text, in the context of the work by the Martin and Seller lab published in eLife. Thus, we did the respective modification in our writing to include the description of their Elife's work, and also compare their conclusions with our conclusions. All these have been updated in our main text, as the following sentences:

“A recent study demonstrated that Myo-II EE form had much weaker motor activity and thus less contractile property (Martin's elife paper). However, with the presence of WT genetic background, overexpression of Myo-II EE form in follicle cells strongly enhanced the formation of stress fibers and also the elongation of egg chambers, while overexpression of Myo-II AA form significantly reduced both (Supplementary Fig. 3g-j). The different effects of Myo-II EE vs. AA forms on ROCK levels, together with on stress fibers and actomyosin contractility, contradicted the hypothesis of the Myo-II mini-filament-mediated ROCK signal amplification.”